# Towards Backwards-Compatible Data with Confounded Domain Adaptation

## Abstract

Most current domain adaptation methods address either covariate shift or label shift, but are not applicable where they occur simultaneously and are confounded with each other. Domain adaptation approaches which do account for such confounding are designed to adapt covariates to optimally predict a particular label whose shift is confounded with covariate shift. In this paper, we instead seek to achieve general-purpose data backwards compatibility. This would allow the adapted covariates to be used for a variety of downstream problems, including on pre-existing prediction models and on data analytics tasks. To do this we consider a modification of generalized label shift (GLS), which we call *confounded shift*. We present a novel framework for this problem, based on minimizing the expected divergence between the source and target conditional distributions, conditioning on possible confounders. Within this framework, we propose using the Gaussian reverse Kullback-Leibler divergence, demonstrating the use of parametric and nonparametric Gaussian estimators of the conditional distribution. We also propose using the Maximum Mean Discrepancy (MMD), introducing a dynamic strategy for choosing the kernel bandwidth, which is applicable even outside the confounded shift setting. Finally, we demonstrate our approach on synthetic and real datasets.

## 1 Introduction

Suppose you have developed a seizure risk prediction model using electroencephalogram (EEG) data, but your hospital lab recently acquired an updated V2 EEG machine. Based on the small amount of data collected for validating the V2 machine, it appears that the V2 machine data distribution is shifted relative to that from the V1 machine. At this point, the problem might appear to call for the use of covariate shift domain adaptation approaches, to adapt the V2 (source) distribution to look like the V1 (target) distribution. Yet additionally, while the V1 dataset comes from a large number of low-risk and high-risk individuals, the V2 dataset thus far is mostly comprised of low-risk volunteers. Ignoring the aforementioned covariate shift problem, this latter problem would seem to fall into the label shift domain adaptation problem setting. Our hypothetical scenario thus combines these two problems: it has both covariate shift and label shift which are confounded with each other.

In the above scenario, the prediction label variable (seizure risk) was coincidentally also the confounder that was correlated with the V1-vs-V2 batch effect. But we might also want to perform statistical analyses for scientific purposes on the EEG data, after combining data from both V1 and V2 machines, to increase our statistical power. For example, we might want to correct for the risk-machine confounding, and then use the adapted EEG data to predict depression, in order to discover EEG features related to depression. Or, using a corrected and combined dataset, we might want to predict EEG data given medication status, to see how certain medications affect EEG features. Our motivating scenario is depicted in Figure 1.

There are multiple obstacles to solving this problem. First, even for tasks that do not involve predicting the confounder (e.g. seizure risk), we cannot simply perform standard covariate-shift domain adaptation, because the source and target datasets should *not* look alike. Second, even for these other tasks that do not involve predicting the confounder (e.g. seizure risk), we cannot assume that the confounder (seizure risk) is

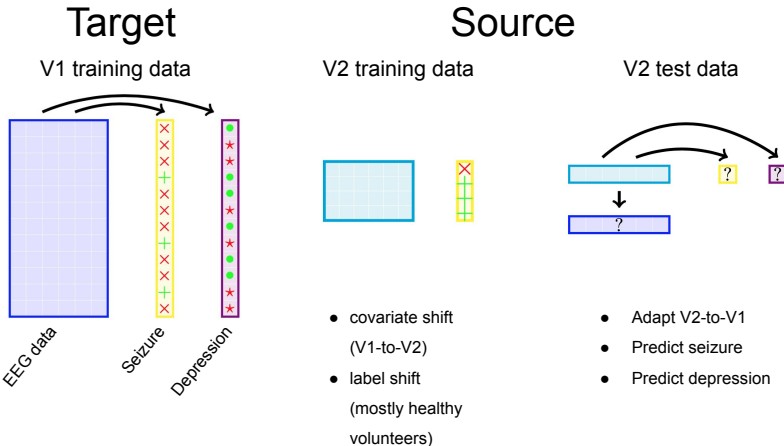

Figure 1: Diagram depicting our motivating scenario.

known for all samples on which we will apply our adaptation. Therefore, we need an adaptation function (e.g. which corrects for the V1-versus-V2 shift) that does not take in the confounder as an input feature (e.g. which does not depend on seizure risk). Third, we cannot discard information unrelated to predicting the confounder. One common approach for domain adaptation is to learn an intermediate representation that is invariant to source-vs-target effects, while still predictive of the label (which is the confounder). But because we want to use adapted-and-combined dataset for a variety of downstream tasks, we need to preserve as much information as possible, not merely the subspace relevant to seizure risk prediction. Fourth, we might have pre-existing prediction models trained on V1 data, which we cannot retrain or finetune on V2 data (either raw or adapted). Thus, in such cases we must make the V2 "backwards-compatible" with models trained on V1 data, producing a V2-to-V1 adapter that is then composed with V1-trained prediction models.

In this paper we seek a domain adaptation method that creates a "general-purpose" fix for the source-vs-target shift in our data, adapting the covariates from V2 to the V1 domain. Then we would be able to combine the V2-to-V1 adapted data with the V1 data, and use them as one domain for a variety of downstream prediction and inference tasks. To begin to address this challenge we assume a modification of generalized label shift (GLS) (Tachet des Combes *et al.*, 2020) which we call *confounded shift*. Confounded shift does not assume that the confounding variable(s) are identically distributed in the source and target domains, or that the covariates are identically distributed in the source and target domains. Rather, it assumes that there exists an adaptation $g : \mathcal{X} \to \mathcal{X}$ from source covariates to target covariates such that the target's conditional distribution of covariates given confounders is equal to that of the adapted-source's conditional distribution. However, we do not assume that the adapted-source's covariates and target's covariates have the same distribution.

In the rest of the paper, we provide a framework for adapting the source to the target, based on minimizing the expected divergence between target and adapted-source conditional distributions, i.e. conditioning on the confounding variables. We show how to compute the expectation with respect to a prior distribution over the confounders, and recommend using an estimator of the product of the source and target confounder distributions. We propose using the Gaussian reverse-KL divergence and the maximum mean discrepancy (MMD) as divergence functions.

Furthermore, using this framework we provide concrete implementations based on the assumption that the source-vs-target batch effect is "simple". In particular, we restrict the adaptation to be affine, or even location-scale (i.e. with a rotation representable by a diagonal matrix). Meanwhile, we consider both simple (e.g. multivariate linear Gaussian) and complex (e.g. Gaussian Process and kernel-based) estimators for the conditional distribution of the covariates given the confounder(s). This assumption is especially intended to adapt structured data, such as biometric sensor outputs, genomic sequencing data, and financial market data,

where domain shifts are typically simple, yet where the input-output mapping is often nonlinear. We are not, in this paper, attempting to adapt an image classification model from photographic inputs to hand-drawn inputs, though we hope our framework could be extended to nonlinear domain adaptation settings.

Software is available at https://github.com/uhhnonymous/anon-submission-tmlr.

## 2 Preliminaries

In this section, we introduce our notation, describe standard approaches to affine-transformation domain adaptation, and provide background on generalized label shift.

### 2.1 Notation

Our notation is inspired by the setting where the confounding variable is the label variable, even though our framework is not strictly intended for this scenario. $\mathcal{X}$ and $\mathcal{Y}$ respectively denote the covariate (input feature) and confounder (output label) space. $X$ and $Y$ denote random variables which take values in $\mathcal{X}$ and $\mathcal{Y}$, respectively. A joint distribution over covariate space $\mathcal{X}$ and confounder space $\mathcal{Y}$ is called a domain $\mathcal{D}$. In our setting, there is a source domain $\mathcal{D}_S$ and a target domain $\mathcal{D}_T$. $\mathcal{D}_S^X, \mathcal{D}_T^X$ denote the marginal distributions of covariates under the source and target domains, respectively; $\mathcal{D}_S^Y, \mathcal{D}_T^Y$ denote the corresponding marginal distributions of confounders. For arbitrary distributions $P$ and $Q$, we assume we have been given a distance or divergence function denoted by $d(P, Q)$. By $\mathcal{N}(\boldsymbol{\mu}, \boldsymbol{\Sigma})$ we denote the Gaussian distribution with mean $\boldsymbol{\mu}$ and covariance $\boldsymbol{\Sigma}$. By $|\cdot|$ we denote the absolute value; by $\det(\cdot)$ we denote the matrix determinant. By $\boldsymbol{A}^\top$ we denote the matrix transpose.

### 2.2 Affine Domain Adaptation based on Gaussian Optimal Transport

Domain adaptation has a closed form affine solution in the special case of two multivariate Gaussian distributions. The optimal transport (OT) map under the type-2 Wasserstein metric for $\boldsymbol{x} \sim \mathcal{N}(\boldsymbol{\mu}_S, \boldsymbol{\Sigma}_S)$ to a different Gaussian distribution $\mathcal{N}(\boldsymbol{\mu}_T, \boldsymbol{\Sigma}_T)$ has been shown (Dowson & Landau, 1982; Knott & Smith, 1984) to be the following:

$$\boldsymbol{x} \mapsto \boldsymbol{\mu}_T + \boldsymbol{A}(\boldsymbol{x} - \boldsymbol{\mu}_S) = \boldsymbol{A}\boldsymbol{x} + (\boldsymbol{\mu}_T - \boldsymbol{A}\boldsymbol{\mu}_S), \tag{1}$$

where

$$\boldsymbol{A} = \boldsymbol{\Sigma}_S^{-1/2} \left( \boldsymbol{\Sigma}_S^{1/2} \boldsymbol{\Sigma}_T \boldsymbol{\Sigma}_S^{1/2} \right)^{1/2} \boldsymbol{\Sigma}_S^{-1/2} = \boldsymbol{A}^\top. \tag{2}$$

This mapping has been applied to a variety of uses (Mallasto & Feragen, 2017; Muzellec & Cuturi, 2018; Shafieezadeh Abadeh *et al.*, 2018; Peyré *et al.*, 2019) in OT and machine learning. For univariate Gaussians $\mathcal{N}(\mu_S, \sigma_S^2)$ and $\mathcal{N}(\mu_T, \sigma_T^2)$, the above transformation simplifies to

$$x \mapsto \mu_T + \frac{\sigma_T}{\sigma_S}(x - \mu_S) = \frac{\sigma_T}{\sigma_S}x + \left(\mu_T - \frac{\sigma_T}{\sigma_S}\mu_S\right). \tag{3}$$

### 2.3 Affine Domain Adaptation Minimizing the Maximum Mean Discrepancy (MMD)

An alternative approach can be derived from representing the distance between target and adapted-source distributions as the distance between mean embeddings. This leads to minimizing the (squared) maximum mean discrepancy (MMD), where the MMD is defined by a feature map $\phi$ mapping features $\boldsymbol{x} \in \mathcal{X}$ to a reproducing kernel Hilbert space $\mathcal{H}$. We denote the feature-space kernel corresponding to $\phi$ as $k_{\mathcal{X}}(\boldsymbol{x}^{(n_1)}, \boldsymbol{x}^{(n_2)}) = \langle \phi(\boldsymbol{x}^{(n_1)}), \phi(\boldsymbol{x}^{(n_2)}) \rangle$. Because the feature-space vectors are assumed to be real, MMD-based adaptation methods typically use the radial basis function (RBF) kernel, which leads to the MMD being zero if and only if the distributions are identical.

If the transformation is affine from source to target, the loss can be written as follows:

$$
\begin{aligned}
\text{MMD}^2(\mathcal{D}_T, \mathcal{D}_S) = & \, \mathbb{E}_{\boldsymbol{x}^{(n_1)}, \boldsymbol{x}^{(n_1)'} \sim \mathcal{D}_T} k_{\mathcal{X}}(\boldsymbol{x}^{(n_1)}, \boldsymbol{x}^{(n_1)'}) \\
& - 2\mathbb{E}_{\boldsymbol{x}^{(n_1)} \sim \mathcal{D}_T, \boldsymbol{x}^{(n_2)} \sim \mathcal{D}_S} k_{\mathcal{X}}(\boldsymbol{x}^{(n_1)}, \boldsymbol{A}\boldsymbol{x}^{(n_2)} + \boldsymbol{b}) \\
& + \mathbb{E}_{\boldsymbol{x}^{(n_2)}, \boldsymbol{x}^{(n_2)'} \sim \mathcal{D}_S} k_{\mathcal{X}}(\boldsymbol{A}\boldsymbol{x}^{(n_2)} + \boldsymbol{b}, \boldsymbol{A}\boldsymbol{x}^{(n_2)'} + \boldsymbol{b}).
\end{aligned} \tag{4}
$$

Prior work has sometimes instead assumed a location-scale transformation (Zhang *et al.*, 2013), or a nonlinear transformation (Liu *et al.*, 2019a). Notably, while previous MMD-based domain adaptation methods have matched feature distributions (Zhang *et al.*, 2013; Liu *et al.*, 2019a; Singh *et al.*, 2020; Yan *et al.*, 2017), joint distributions of features and label (Long *et al.*, 2013), or the conditional distribution of label given features (Long *et al.*, 2013), they have generally not considered matching the conditional distribution of features given labels. One exception to this is IWCDAN (Tachet des Combes *et al.*, 2020), which however aligns datasets via sample importance weighting rather than a feature-space transformation.

Despite their theoretical attractiveness, MMD-based domain adaptation methods tend to struggle in practice, such as on single-cell genomics data (Singh *et al.*, 2020), for a few reasons. First, because MMD is a non-convex functional, it tends get stuck in local minima. This related to another practical weakness, which is that it is very sensitive to the choice of length-scale / bandwidth hyperparameter. When the bandwidth is too small, each datapoint are seen as dissimilar to all other points except itself. If the source and target data are separated, the second term in Eq. (4) will be approximately zero with vanishing gradient far from a skinny Gaussian, so no progress will be made. Yet when the bandwidth is too large, the gradient also vanishes with different datapoints together at the flat top of a wide Gaussian. Various measures have been proposed for these problems, such as adding a discriminative term to the objective (Wang *et al.*, 2020) and choosing the (fixed) bandwidth in a data-driven way from the entire dataset (Singh *et al.*, 2020).

### 2.4 Background on Covariate Shift, Label Shift, and Generalized Label Shift

Domain adaptation methods typically assume either covariate shift or label shift. With covariate shift, the marginal distribution over covariates differs between source and target domains. However, for any particular covariate, the conditional distribution of the label given the covariate is identical between source and target. With label shift, the marginal distribution over labels differs between source and target domains. However, for any particular label, the conditional distribution of the covariates given the label is identical between source and target domain.

More recently, generalized label shift was introduced to allow covariate distributions to differ between source and target domains (Tachet des Combes *et al.*, 2020). Generalized label shift (GLS) instead assumes that, given a transformation function $Z = g(X)$ applied to inputs from both source and target domains, the conditional distributions of $Z$ given $Y = y$ are identical for all $y$. This is a weak assumption, and it applies to our problem setting as well. However, it is designed for the scenario where we simply need $g$ to preserve information only for predicting $Y$ given $X \sim \mathcal{D}_S^X$.

## 3 Confounded Domain Adaptation

For the time being, we will consider our motivating scenario in which our ultimate goal is to reuse a minimum-risk binary classification hypothesis $h : \mathcal{X} \to \{0, 1\}$ in a new deployment setting. We treat the deployment setting as the source domain, instead of (as is typical in domain adaptation) the target domain. And instead of learning an end-to-end predictor for the deployment domain, we learn an adaptation $g$ from it to the target domain for which we have a large number of labelled examples. Then, to perform predictions on the deployment (source) domain, we first adapt them to the target domain, and then we apply the prediction model trained on the target domain. In other words, we do not need to retrain $h$, and instead apply $h \circ g$ to incoming unlabelled source samples. Similarly, other prediction tasks and statistical analyses can be identically applied to target domain data and adapted-source domain data.

In many real-world structured data applications, new data sources are designed with "backwards-compatibility" in mind, with the goal that updated sensor and assays provide at least as much information as the earlier

versions. We assume the existence of a "true" noise-free mapping $g$ from the deployment domain to the large labelled dataset domain. We further assume that this mapping is affine, i.e., $g(\boldsymbol{x}) = \boldsymbol{Ax} + \boldsymbol{b}$ for some $\boldsymbol{A}, \boldsymbol{b}$ .

The algorithms developed under our framework could instead be applied when treating the deployment setting as the target domain and the labelled dataset as the source domain. This would entail retraining $h$ on adapted data, and then applying $h$ to new samples. However, such usage is not the focus of this paper.

We assume $N_S$ and $N_T$ samples from the source and target domain, respectively. We assume each sample has feature vector $\boldsymbol{x}^{(n)} \in \mathbb{R}^M$. Each sample has confounding variables represented as $\boldsymbol{y}^{(n)}$, which could be categorical, continuous, a concatenation of both, or even a more general object such as a string. The confounders will (unless otherwise indicated) be accessed via a user-specified confounder-space kernel function $k_{\mathcal{Y}}(\boldsymbol{y}^{(n_1)}, \boldsymbol{y}^{(n_2)})$.

### 3.1 Our Assumption: Confounded Shift

In our case, given $X \sim \mathcal{D}_S^X$, we instead want to recover what it would have been had we observed the same object from the data generating process corresponding to the target domain $X \sim \mathcal{D}_T^X$. In other words, the mapping $g(X)$ should not only preserve information in $X$ useful for predicting $Y$, but ideally all information in $X \sim \mathcal{D}_S^X$ that is contained in $X \sim \mathcal{D}_T^X$.

**Relation to Generalized Label Shift** Suppose GLS intermediate representation $g(X)$ were extended to be a function of both $X$ and an indicator variable $D$ specifying whether a sample is taken from the target or the source domain. Then, given this extended representation $\{X, D\}$, we restrict $\tilde{g}(\{X, D\})$ as follows,

$$\tilde{g}(\{X, D\}) = \begin{cases} g(X) & D = S \\ X & D = T \end{cases} \tag{5}$$

so that samples from the source distribution are adapted by $g(\cdot)$, while those from the target distribution pass through unchanged. With this extended representation, as well as the restriction on $\tilde{g}$, confounded shift and GLS coincide. Note that while confounded shift is stronger than GLS, both allow $\mathcal{D}_S^X \neq \mathcal{D}_T^X$; and just as GLS allows $\mathcal{D}_S^{g(X)} \neq \mathcal{D}_T^{g(X)}$, we analogously allow $\mathcal{D}_S^{g(X)} \neq \mathcal{D}_T^X$.

**Graphical representation** We may formulate our setting with latent variables $Z$ corresponding to features before the target-to-source (e.g. V1-to-V2) update. To do this, we assume that $g$ (source-to-target) is invertible with inverse $g^{-1}$ (target-to-source). We may depict our assumption with the following graphical model

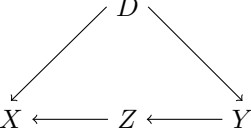

where latent features $Z$ always follow the target domain distribution $p(Z = z | Y = y) = p_{\mathcal{D}_T}(X = z | Y = y)$, while observed features follow

$$p(X = x | D, Z = z) = \begin{cases} \delta(x - g^{-1}(z)) & D = S \\ \delta(x - z) & D = T \end{cases} \tag{6}$$

where $\delta$ is the Dirac delta. By inspection of the graphical model, our setting is a combination of *prior probability shift* and *covariate observation shift* as defined in (Kull & Flach, 2014). Note that latent features $Z$ are generated from confounders $Y$, which suggests that we use a generative model for domain adaptation.

The previous assumptions as well as our confounded shift assumption are summarized in Table 1.

### 3.2 Main Idea

Our primary aim is to infer a transformation that is broadly applicable, so given observed source domain features $X = x$ we will seek to reconstruct $z$ with minimal error. Our secondary aim is to minimize error

Table 1: Domain adaptation settings

| Name | Shift | Assumed Invariant |
|---|---|---|
| Covariate Shift | $\mathcal{D}_S^X \neq \mathcal{D}_T^X$ | $\forall x \in \mathcal{X}, \mathcal{D}_S(Y|X = x) = \mathcal{D}_T(Y|X = x)$ |
| Label Shift | $\mathcal{D}_S^Y \neq \mathcal{D}_T^Y$ | $\forall y \in \mathcal{Y}, \mathcal{D}_S(X|Y = y) = \mathcal{D}_T(X|Y = y)$ |
| Generalized Label Shift | $\mathcal{D}_S^Y \neq \mathcal{D}_T^Y$ | $\forall y \in \mathcal{Y}, \mathcal{D}_S(g(X)|Y = y) = \mathcal{D}_T(g(X)|Y = y)$ |
| **Confounded Shift** | $\mathcal{D}_S^Y \neq \mathcal{D}_T^Y$ | $\forall y \in \mathcal{Y}, \mathcal{D}_S(g(X)|Y = y) = \mathcal{D}_T(X|Y = y)$ |

on downstream prediction tasks, which for simplicity is assumed to be binary classification. Formally, the hypothesis is a fixed binary classification function $h : \mathcal{X} \to \{0, 1\}$. We seek to choose $\hat{g}$ which minimizes the accuracy loss induced (by unknown shift $g^{-1}$) on hypothesis $h$ under distribution $\mathcal{D}_T^X$:

$$p_{\mathcal{D}_T^X}\Big(h \circ \hat{g}\big(g^{-1}(X)\big) \neq h(X)\Big). \tag{7}$$

Contrary to typical domain adaptation settings, we expect that $N_S < N_T$, since we are adapting $N_S$ datapoints from the new V2 sensor, from which we have few samples. We expect an abundance of prediction labels on our target dataset (i.e. from the V1 sensor), though this is irrelevant since $h$ is already trained and fixed.

Our proposal, which we dub ConDo, is to minimize the expected distance (or divergence) $d$ between the conditional distributions of source and target given confounders, under some specified prior distribution over the confounders. Our goal is to find the optimal linear transformation $g(\boldsymbol{x}) = \boldsymbol{A}\boldsymbol{x} + \boldsymbol{b}$ of the source to target, leading to the following objective:

$$\min_{\boldsymbol{A}, \boldsymbol{b}} \mathbb{E}_{y \sim \mathcal{D}_{\text{prior}}^Y} \ d\Big(\mathcal{D}_T(\boldsymbol{x}|Y = y), \mathcal{D}_S(\boldsymbol{A}\boldsymbol{x} + \boldsymbol{b}|Y = y)\Big). \tag{8}$$

In certain scenarios, particularly scientific analyses, it is important for explainability that each $i$th adapted feature $[\boldsymbol{A}\boldsymbol{x}^{(n)} + \boldsymbol{b}]_i$ be derived only from the original feature $[\boldsymbol{x}^{(n)}]_i$. So we will examine both full affine transformations and also transformations where $\boldsymbol{A}$ is restricted to be diagonal $\boldsymbol{A} = \text{diag}(\boldsymbol{a})$; the latter is sometimes referred to as a location-scale adaptation (Zhang *et al.*, 2013).

Three ingredients remain to turn this framework into a concrete algorithm: the choice of prior confounder distribution $\mathcal{D}_{\text{prior}}^Y$ (Section 3.3), the distance/divergence function $d$ (Section 3.4), and the conditional distribution estimators $\mathcal{D}.(\cdot|Y = y)$ (Section 3.5).

### 3.3 Choice of Confounder Prior Distribution

The appropriate choice of confounder prior depends upon two considerations. Firstly, all things else being equal, it would be best for this prior to match the distribution over the confounder(s) that we expect to see in the future. Our approach minimizes risk under the chosen prior distribution, which suggests choosing this prior to match the deployment distribution. For example, if the primary downstream task is to predict the confounding variable on future incoming samples, and the confounder's distribution on the target dataset is representative of future samples, then this suggests choosing $\mathcal{D}_{\text{prior}}^Y := \hat{\mathcal{D}}_S^Y$, the empirical distribution of $Y$ in our source dataset. However, there is a second consideration which may override the above logic: our conditional distribution estimators may be poor extrapolators, and so we should minimize the distance between the conditional distributions only where we can estimate both with high accuracy. This suggests choosing to perform minimization over confounder values that are likely under both source $\mathcal{D}_S^Y$ and target $\mathcal{D}_T^Y$ distributions, thus estimating the product of the two distributions. On the other hand, if the conditional distribution admits easy extrapolation, then it may be appropriate to minimize over values that are likely in either source $\mathcal{D}_S^Y$ or target $\mathcal{D}_T^Y$, thus summing the two distributions.

Formalizing the above reasoning, we define four possible choices of the confounder prior. We will let each prior have non-negative support over the union of confounder values in the source and target datasets, so

that each can be represented as probabilistic weights attached to each sample. The source, target, and sum priors can be trivially represented as follows:

$$\hat{\mathcal{D}}_S^Y := \frac{1}{N_S} \sum_n^{N_S} \delta(y - Y_S^{(n)}), \tag{9}$$

$$\hat{\mathcal{D}}_T^Y := \frac{1}{N_T} \sum_n^{N_T} \delta(y - Y_T^{(n)}), \tag{10}$$

$$\hat{\mathcal{D}}_+^Y := \frac{1}{2N_S} \sum_n^{N_S} \delta(y - Y_S^{(n)}) + \frac{1}{2N_T} \sum_n^{N_T} \delta(y - Y_T^{(n)}). \tag{11}$$

where $\delta$ is the Dirac delta function. For the product prior, we note that the empirical distributions may have non-intersecting support. Without smoothing the priors, this would be problematic because (for example) the weight $\boldsymbol{w}_S^{(n)}$ attached to a particular sample $Y_S^{(n)}$ would be zero unless $Y_S^{(n)} = Y_T^{(i)}$ for some $1 \le i \le N_T$. This happens almost never if $Y$ comes from a continuous distribution. Therefore, before computing their product, we smooth the priors using the confounder-space kernel $k_{\mathcal{Y}}$ as follows:

$$\hat{\mathcal{D}}_*^Y := \sum_n^{N_S} \boldsymbol{w}_S^{(n)} \delta(y - Y_S^{(n)}) + \sum_n^{N_T} \boldsymbol{w}_T^{(n)} \delta(y - Y_T^{(n)}), \text{ where} \tag{12}$$

$$\boldsymbol{w}_S^{(n)} \propto \left( \frac{\sum_{i=1}^{N_S} k_{\mathcal{Y}}(Y_S^{(i)}, Y_S^{(n)})}{\sum_{j=1}^{N_S} \sum_{i=1}^{N_S} k_{\mathcal{Y}}(Y_S^{(i)}, Y_S^{(j)})} \times \frac{\sum_{i=1}^{N_T} k_{\mathcal{Y}}(Y_T^{(i)}, Y_S^{(n)})}{\sum_{j=1}^{N_T} \sum_{i=1}^{N_T} k_{\mathcal{Y}}(Y_T^{(i)}, Y_T^{(j)})} \right)^{1/2} \tag{13}$$

$$\boldsymbol{w}_T^{(n)} \propto \left( \frac{\sum_{i=1}^{N_S} k_{\mathcal{Y}}(Y_S^{(i)}, Y_T^{(n)})}{\sum_{j=1}^{N_S} \sum_{i=1}^{N_S} k_{\mathcal{Y}}(Y_S^{(i)}, Y_S^{(j)})} \times \frac{\sum_{i=1}^{N_T} k_{\mathcal{Y}}(Y_T^{(i)}, Y_T^{(n)})}{\sum_{j=1}^{N_T} \sum_{i=1}^{N_T} k_{\mathcal{Y}}(Y_T^{(i)}, Y_T^{(j)})} \right)^{1/2}. \tag{14}$$

where $\boldsymbol{w}_S$ and $\boldsymbol{w}_T$ are normalized so $\sum_n \boldsymbol{w}_S^{(n)} + \sum_n \boldsymbol{w}_T^{(n)} = 1$. The product prior $\hat{\mathcal{D}}_*^Y$ is the most conservative choice, so we recommend it as the default, and use it for all experiments in this paper.

### 3.4 Conditional Distribution Distance/Divergence Function

Below, we propose using the reverse-KL divergence and the MMD in our loss function. Both yield simple, efficient algorithms, including a closed-form solution for the reverse KL divergence with location-scale adaptation. Note that, as we discuss in Future Work, other divergences are possible within our framework. In particular, OT-based distances are likely to offer higher accuracy at greater computational expense. However, we limit ourselves to these two divergences for their low computational cost, and to focus on the overall proposed framework rather than the computational challenges and opportunities that arise from combining ConDo with OT.

#### 3.4.1 Reverse KL Divergence under Gaussianity

It can be straightforwardly shown that the linear map Eq. (1) derived from optimal transport leads to adapted data being distributed according to the target distribution. That is, $\boldsymbol{\mu}_P + \boldsymbol{A}(\boldsymbol{x} - \boldsymbol{\mu}_Q) \sim \mathcal{N}(\boldsymbol{\mu}_P, \boldsymbol{\Sigma}_P)$. Therefore, the KL-divergence from the target distribution to the adapted source data distribution is minimized to 0, and similarly for the KL-divergence from the adapted source data distribution to the target distribution. This motivates using the KL-divergence as a loss function, with either the forward KL-divergence $d(P, Q) := d_{KL}(P||Q)$ or reverse KL-divergence $d(P, Q) := d_{KL}(Q||P)$. While the forward KL-divergence from target to adapted-source appears to be the natural choice, we instead propose to use the reverse KL-divergence. Due to its computational tract and well-conditioned, the reverse KL has found wide use in variational inference (Blei et al., 2017) and reinforcement learning (Kappen et al., 2012; Levine, 2018). We will show that it has similar benefits in domain adaptation.

In either case, it can be shown that minimizing Eq. (8) requires estimating the conditional means and conditional covariances, according to both the source and target domain estimators, evaluated at each

$y \sim \mathcal{D}_{\text{prior}}^Y$. (If the transformation is location-scale rather than full affine, KL divergence minimization requires only the conditional variances for each feature.)

Given $N$ samples in the prior distribution, each with weight given by $\boldsymbol{w}_n, 1 \le n \le N$, let the source and target estimated conditional means be given by $\boldsymbol{\mu}_S^{(n)}, \boldsymbol{\mu}_T^{(n)}$, and the conditional covariances be given by $\boldsymbol{\Sigma}_S^{(n)}, \boldsymbol{\Sigma}_T^{(n)}$, respectively.

For the forward-KL divergence, this leads to the following objective:

$$\min_{\boldsymbol{A},\boldsymbol{b}} 2 \log\left(|\det(\boldsymbol{A})|\right) + \sum_{n=1}^N \boldsymbol{w}_n * \left[\text{tr}\left(\left[\boldsymbol{A}\boldsymbol{\Sigma}_S^{(n)}\boldsymbol{A}^\top\right]^{-1}\boldsymbol{\Sigma}_T^{(n)}\right)\right.$$
$$\left. + \left(\boldsymbol{A}\boldsymbol{\mu}_S^{(n)} + \boldsymbol{b} - \boldsymbol{\mu}_T^{(n)}\right)^\top \left[\boldsymbol{A}\boldsymbol{\Sigma}_S^{(n)}\boldsymbol{A}^\top\right]^{-1}\left(\boldsymbol{A}\boldsymbol{\mu}_S^{(n)} + \boldsymbol{b} - \boldsymbol{\mu}_T^{(n)}\right)\right]. \quad (15)$$

Meanwhile, for the reverse-KL divergence, we instead have:

$$\min_{\boldsymbol{A},\boldsymbol{b}} -2 \log\left(|\det(\boldsymbol{A})|\right) + \sum_{n=1}^N \boldsymbol{w}_n * \left[\text{tr}\left(\boldsymbol{\Sigma}_T^{(n)^{-1}}\boldsymbol{A}\boldsymbol{\Sigma}_S^{(n)}\boldsymbol{A}^\top\right)\right.$$
$$\left. + \left(\boldsymbol{A}\boldsymbol{\mu}_S^{(n)} + \boldsymbol{b} - \boldsymbol{\mu}_T^{(n)}\right)^\top \boldsymbol{\Sigma}_T^{(n)^{-1}}\left(\boldsymbol{A}\boldsymbol{\mu}_S^{(n)} + \boldsymbol{b} - \boldsymbol{\mu}_T^{(n)}\right)\right]. \quad (16)$$

Besides being more efficient to optimize (requiring matrix inversion once rather than at each iteration), the reverse-KL objective minimizes the negative log-abs-determinant of $\boldsymbol{A}$, which functions as a log-barrier away from 0, maintaining the same sign of the determinant across optimization iterations. This is useful, because the linear mapping between two Gaussians is not unique. The reverse-KL divergence, combined with an initial iterate (e.g. the identity matrix) with a positive determinant, chooses the mapping which preserves rather than reverses the orientation. In contrast, the forward-KL objective is liable to produce iterates with oscillating signs of $\det(\boldsymbol{A})$.

That the $(-\log|\det(\boldsymbol{A})|)$ term arises naturally out of the reverse-KL divergence is of potential independent interest. Preventing collapse into trivial solutions is a known problem with MMD-based domain adaptation (Singh *et al.*, 2020; Wu *et al.*, 2021). The reverse-KL objective may inspire a new regularization penalty for this problem. The log-det heuristic was previously proposed (Fazel *et al.*, 2003) as a smooth concave surrogate for matrix rank minimization, while here it prevents rank collapse.

Furthermore, in the case of a location-scale adaptation, the reverse-KL divergence can be obtained via a fast exact closed-form solution. Further details are given in Appendix A.

### 3.4.2 The Conditional Maximum Mean Discrepancy

We extend MMD-based domain adaptation to match conditional distributions by sampling from the prior confounder distribution. For a particular $y \in \mathcal{Y}$ sampled from the prior, suppose we have a way of sampling from $\mathcal{D}_T(\cdot|Y = y)$ and $\mathcal{D}_S(\cdot|Y = y)$. Then, we have

$$d\left(\mathcal{D}_T(\cdot|Y = y), \mathcal{D}_S(\cdot|Y = y)\right) := \text{MMD}^2(\mathcal{D}_T(\cdot|Y = y), \mathcal{D}_S(\cdot|Y = y)) \quad (17)$$

$$= \mathbb{E}_{\boldsymbol{x}^{(n_1)}, \boldsymbol{x}^{(n_1)'} \sim \mathcal{D}_T(\cdot|Y=y)} k_\mathcal{X}\left(\boldsymbol{x}^{(n_1)}, \boldsymbol{x}^{(n_1)'}\right)$$
$$- 2\mathbb{E}_{\boldsymbol{x}^{(n_1)} \sim \mathcal{D}_T(\cdot|Y=y), \boldsymbol{x}^{(n_2)} \sim \mathcal{D}_S(\cdot|Y=y)} k_\mathcal{X}\left(\boldsymbol{x}^{(n_1)}, \boldsymbol{A}\boldsymbol{x}^{(n_2)} + \boldsymbol{b}\right)$$
$$+ \mathbb{E}_{\boldsymbol{x}^{(n_2)}, \boldsymbol{x}^{(n_2)'} \sim \mathcal{D}_S(\cdot|Y=y)} k_\mathcal{X}\left(\boldsymbol{A}\boldsymbol{x}^{(n_2)} + \boldsymbol{b}, \boldsymbol{A}\boldsymbol{x}^{(n_2)'} + \boldsymbol{b}\right). \quad (18)$$

We efficiently minimize this objective by sampling batches from the conditional distributions, combined with (batch) gradient descent with momentum. To sample from the conditional distributions, we sample (with

replacement) from the empirical distributions, with sample weights derived from the confounder-space kernel $k_\mathcal{Y}$. This is described in more detail in Section 3.5.4.

Furthermore, we propose to dynamically recompute the bandwidth for each batch during the optimization procedure. As our algorithm adapts the source to the target, our bandwidth estimate will progressively update to continue focusing on matching the source and target. Given source sample $\boldsymbol{X}^S \in \mathbb{R}^{N_{\text{batch}} \times M}$ and target sample $\boldsymbol{X}^T \in \mathbb{R}^{N_{\text{batch}} \times M}$ and the current transformation parameters $(\boldsymbol{A}, \boldsymbol{b})$, the squared-bandwidth for a single dimension $i$ of the features is computed as follows:

$$\sigma_i^2 = \frac{1}{N_{\text{batch}}} \sum_n^{N_{\text{batch}}} \left( \boldsymbol{X}_{n,i}^T - ([\boldsymbol{X}^S \boldsymbol{A}^\top]_{n,i} + \boldsymbol{b}_i) \right)^2. \tag{19}$$

## 3.5 Estimators for the Conditional Distribution

In this section, we present four estimators for $\mathcal{D}_\cdot(\boldsymbol{x}|Y = y)$. The first three are designed to accompany the KL-divergence, estimating conditional means $\boldsymbol{\mu}_S^{(n)}, \boldsymbol{\mu}_T^{(n)}$ and conditional covariances $\boldsymbol{\Sigma}_S^{(n)}, \boldsymbol{\Sigma}_T^{(n)}$ given each sample $y$ from the confounder prior. The final estimator (to be used in conjunction with MMD) allows us to sample a batch of $\boldsymbol{x}$s for each given $y$ from the confounder prior.

### 3.5.1 Linear Gaussian Distribution for KL-divergence Minimization

Here we model $\boldsymbol{x}$ conditioned on real-valued vector $\boldsymbol{y}$ as linear Gaussian:

$$\mathcal{D}_\cdot(\boldsymbol{x}|Y = \boldsymbol{y}) = \mathcal{N}(\boldsymbol{B}\boldsymbol{y}, \boldsymbol{\Lambda}^{-1}). \tag{20}$$

Because $\boldsymbol{x}$ is potentially high-dimensional, we estimate parameters $(\boldsymbol{B}, \boldsymbol{\Lambda})$ with regularized multivariate linear regression and the Graphical Lasso (Friedman *et al.*, 2008). This model is homoscedastic, because all samples in the source dataset will have identical estimated covariances $\boldsymbol{\Sigma}_S^{(n)} = \boldsymbol{\Lambda}_S^{-1}, \boldsymbol{\Sigma}_T^{(n)} = \boldsymbol{\Lambda}_T^{-1}, \forall 1 \le n \le N$. Meanwhile, we will obtain different predicted means for each source and target sample $\boldsymbol{\mu}_S^{(n)}, \boldsymbol{\mu}_T^{(n)}$. This estimator requires confounder $y$ to be quantitative; we use one-hot encoding to convert categorical confounders.

### 3.5.2 Product of Gaussian Mixture Models for KL-divergence Minimization

Assume for the moment that confounder $Y$ is univariate categorical. For each observed value $Y = y$, we estimate the conditional means $\boldsymbol{\mu}_S|Y = y, \boldsymbol{\mu}_T|Y = y$ from the empirical means and the conditional covariances $\boldsymbol{\Sigma}_S|Y = y, \boldsymbol{\Sigma}_T|Y = y$ from the empirical covariances using Graphical Lasso (Friedman *et al.*, 2008). To handle multivariate categorical $Y$ over $J$ categorical variables, we take the product of each conditional distribution, which (after normalizing) is itself Gaussian. Formally, consider the feature-confounder pair $(\boldsymbol{x}^{(n)}, \boldsymbol{y}^{(n)})$ for a particular sample $n$, where $\boldsymbol{x}^{(n)} \in \mathbb{R}^M$ and $\boldsymbol{y}^{(n)} \in \mathcal{C} = \mathcal{C}_1 \times \ldots \times \mathcal{C}_j \times \ldots \times \mathcal{C}_J$, with $\mathcal{C}_j = \{1, \ldots, k, \ldots, K\}$, where each categorical variable is $K$-ary. The conditional means/precisions are indexed as $(\boldsymbol{\mu}_k^{(j)}, \boldsymbol{\Lambda}_k^{(j)})$. Our product-of-experts assumption yields

$$\mathcal{D}_\cdot(\boldsymbol{x}^{(n)}|\boldsymbol{y}^{(n)}) = \prod_{j=1}^J p_\cdot(\boldsymbol{x}^{(n)}|\boldsymbol{y}_j^{(n)}) \propto \prod_{j=1}^J \mathcal{N}\left( \boldsymbol{\mu}_{\boldsymbol{y}_j^{(n)}}^{(j)}, \left[ \boldsymbol{\Lambda}_{\boldsymbol{y}_j^{(n)}}^{(j)} \right]^{-1} \right) \tag{21}$$

$$= \mathcal{N}\left( \left[ \sum_{j=1}^J \boldsymbol{\Lambda}_{\boldsymbol{y}_j^{(n)}}^{(j)} \right]^{-1} \left( \sum_{j=1}^J \boldsymbol{\Lambda}_{\boldsymbol{y}_j^{(n)}}^{(j)} \boldsymbol{\mu}_{\boldsymbol{y}_j^{(n)}}^{(j)} \right), \left[ \sum_{j=1}^J \boldsymbol{\Lambda}_{\boldsymbol{y}_j^{(n)}}^{(j)} \right]^{-1} \right). \tag{22}$$

This estimator requires confounder $y$ to be categorical (though potentially multivariate); we use KMeans clustering to quantize each continuous confounding variable into categories.

### 3.5.3 Univariate Gaussian Process for KL-divergence Minimization

We will only use this model for location-scale transformations, so we model each feature independently. Without loss of generality, for feature $i$ in source domain $\mathcal{D}_S(\cdot|Y = y)$, we have $\boldsymbol{x}_i \in \mathbb{R}^{N_S}$ modeled using a

Gaussian process (GP):

$$f(y) \sim \mathcal{GP}(m(y), k_{\mathcal{Y}}(y, y')). \tag{23}$$

Having fit the GP on the source dataset $\boldsymbol{X}_i^S \mapsto Y^S$, we evaluate it to compute and store the conditional mean and variance for each $y$ taken from the confounder prior. This process is repeated for all features, on both the source and target datasets.

### 3.5.4 Conditional Distribution Sampling for MMD Minimization

We model each of the conditional distributions $\mathcal{D}_T(\boldsymbol{x}|Y = y)$ and $\mathcal{D}_S(\boldsymbol{x}|Y = y)$ using Nadaraya-Watson kernel regression (Nadaraya, 1964; Watson, 1964). For each observed value of $Y = y$, we compute dataset sample weight $w(y^{(n)}) \propto k_{\mathcal{Y}}(y, y^{(n)})$ for all samples in the target dataset and source dataset, respectively. Then, we sample (with repeats) from this distribution. For example, to sample from the source conditional $\mathcal{D}_S(\cdot|Y = y)$ for a given $y$, we assign each source sample $X_S^{(n)}$ a weight proportional to $k_{\mathcal{Y}}(Y^{(n)}, y)$.

**Additional Implementation Details**   The design of confounder-space kernels is discussed in Appendix B. Computational speedups for categorical confounders are discussed in Appendix C.

## 4   Experiments

We analyze our approach on synthetic data in Section 4.1 and on real data in Section 4.2. We also include experiments validating our dynamic kernel bandwidth strategy for MMD in Appendix D.1.

### 4.1   Synthetic Data

### 4.1.1   1d Data with 1d Continuous Confounder

We first examine confounded domain adaptation in the context of a single-dimensional feature confounded by a single-dimensional continuous confounder. Our results are illustrated in Figure 2. We analyze the performance of vanilla and ConDo adaptations, when the effect of the continuous confounder is linear homoscedastic (left column), linear heteroscedastic (middle column), and nonlinear heteroscedastic (right column). In all cases, there is confounded shift because in the target domain the confounder is uniformly distributed in $(0, 8)$, while for the source domain in $(4, 8)$.

We repeat the above experimental setup, but with modifications to verify whether our approach can be accurate even when its assumptions no longer apply. We run experiments with and without noisy batch effects, with-vs-without label shift (i.e. different distributions over the confounder between source and target), and with-vs-without feature shift (i.e. with and without batch effect), for a total of 8 shift-type settings.

Overall, we find that ConDo is robust to violations of the confounded shift assumption. We find that adding noise to the batch effect does not affect the performance of ConDo. We also find that without label shift (when confounding-awareness is unnecessary) ConDo is non-inferior to confounding-unaware methods And, without feature-shift (when the true transformation is the identity, even if label shift makes the marginal feature distributions differ), we find that only ConDo reliably chooses approximately the identity transformation. The noise-free and noisy results are summarized in Tables 2 and 3, respectively. The full results are provided in Appendix D.2.

### 4.1.2   1d Data with Multiple Continuous Confounders

We extend the previous experiment to consider the scalability of ConDo to multidimensional confounders. For the noise-free, label-shift, feature-shift setting, keeping all other experimental settings and hyperparameters identical, we vary the number of confounders from 1 to 32. We first augment the number of confounders by appending additional irrelevant "confounders", sampled from $\mathcal{N}(0, 1)$, to our inputs to the ConDo method. The results, shown in Figure 3, indicate that ConDo Linear Reverse-KL and GP Reverse-KL barely increase in rMSE and maintain superiority for up to 32 confounders. ConDo PoGMM Reverse-KL gradually worsens,

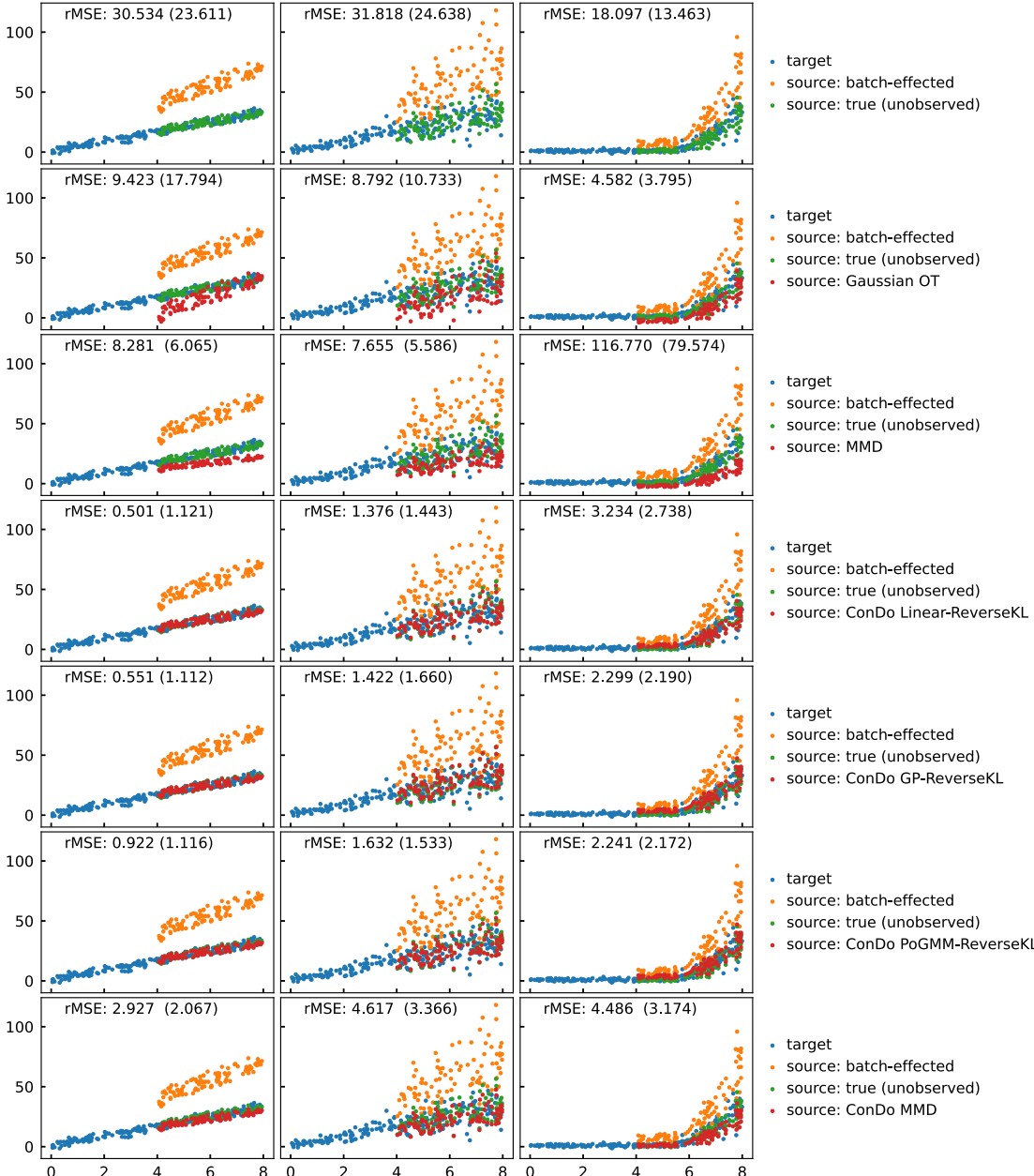

Figure 2: ConDo methods are superior to Gaussian OT when confounded label shift and feature shift are present (with no noise added after shifting). The columns, in order, correspond to a confounder with a linear homoscedastic effect, a confounder with a linear heteroscedastic effect, and a confounder with a nonlinear heteroscedastic effect. The first row depicts the problem setup, while the remaining rows depict the performance of Gaussian OT and our ConDo methods. Red points overlapping with green points is indicative of high accuracy. In each subplot, we provide the rMSE on training source data (depicted), and in parentheses, the rMSE on heldout source data (not depicted) generated with confounder sampled from target prior $\mathcal{D}_T^Y$. The printed rMSEs are averaged over 5 independent random simulation runs, while the plots depict the results from the final simulation run.

Table 2: Summary of results for 1d data with 1d continuous confounder without noise

| | No Noise, Label-Shifted and Feature-Shifted | | |
| --- | --- | --- | --- |
| | Homoscedastic Linear | Heteroscedastic Linear | Nonlinear |
| Before Correction | 30.534 (23.611) | 31.818 (24.638) | 18.097 (13.463) |
| Oracle | 0.000 (0.000) | 0.000 (0.000) | 0.000 (0.000) |
| Gaussian OT | 9.423 (17.794) | 8.792 (10.733) | 4.582 (3.795) |
| MMD | 8.281 (6.065) | 7.655 (5.586) | 116.770 (79.574) |
| ConDo Linear-ReverseKL | 0.501 (1.121) | 1.376 (1.443) | 3.234 (2.738) |
| ConDo GP-ReverseKL | 0.551 (1.112) | 1.422 (1.660) | 2.299 (2.190) |
| ConDo PoGMM-ReverseKL | 0.922 (1.116) | 1.632 (1.533) | 2.241 (2.172) |
| ConDo MMD | 2.927 (2.067) | 4.617 (3.366) | 4.486 (3.174) |
| | No Noise, Label-Shifted Only | | |
| | Homoscedastic Linear | Heteroscedastic Linear | Nonlinear |
| Before Correction | 0.000 (0.000) | 0.000 (0.000) | 0.000 (0.000) |
| Oracle | 0.000 (0.000) | 0.000 (0.000) | 0.000 (0.000) |
| Gaussian OT | 9.638 (16.972) | 7.496 (9.569) | 4.058 (3.470) |
| MMD | 8.804 (6.823) | 7.483 (5.803) | 5.960 (4.817) |
| ConDo Linear-ReverseKL | 0.206 (0.158) | 0.718 (0.552) | 3.437 (2.841) |
| ConDo GP-ReverseKL | 0.194 (0.148) | 0.719 (0.550) | 1.236 (1.053) |
| ConDo PoGMM-ReverseKL | 0.628 (0.475) | 0.855 (0.638) | 0.861 (0.721) |
| ConDo MMD | 2.794 (2.254) | 2.949 (2.280) | 0.000 (0.000) |
| | No Noise, Feature-Shifted Only | | |
| | Homoscedastic Linear | Heteroscedastic Linear | Nonlinear |
| Before Correction | 24.325 (24.455) | 22.504 (24.451) | 14.020 (13.445) |
| Oracle | 0.000 (0.000) | 0.000 (0.000) | 0.000 (0.000) |
| Gaussian OT | 1.273 (1.258) | 2.045 (2.139) | 2.017 (1.931) |
| MMD | 3.307 (3.223) | 2.410 (2.500) | 3.677 (3.691) |
| ConDo Linear-ReverseKL | 1.130 (1.104) | 1.597 (1.520) | 2.596 (2.508) |
| ConDo GP-ReverseKL | 1.138 (1.114) | 1.726 (1.651) | 2.135 (2.091) |
| ConDo PoGMM-ReverseKL | 1.158 (1.134) | 1.466 (1.417) | 2.361 (2.234) |
| ConDo MMD | 1.218 (1.162) | 1.327 (1.264) | 2.671 (2.476) |
| | No Noise, No Shift | | |
| | Homoscedastic Linear | Heteroscedastic Linear | Nonlinear |
| Before Correction | 0.000 (0.000) | 0.000 (0.000) | 0.000 (0.000) |
| Oracle | 0.000 (0.000) | 0.000 (0.000) | 0.000 (0.000) |
| Gaussian OT | 1.044 (1.095) | 1.328 (1.307) | 1.358 (1.231) |
| MMD | 0.971 (1.001) | 1.348 (1.343) | 1.505 (1.218) |
| ConDo Linear-ReverseKL | 0.192 (0.193) | 0.737 (0.709) | 1.147 (1.053) |
| ConDo GP-ReverseKL | 0.182 (0.183) | 0.688 (0.669) | 0.792 (0.732) |
| ConDo PoGMM-ReverseKL | 0.124 (0.128) | 0.747 (0.726) | 1.386 (1.364) |
| ConDo MMD | 0.222 (0.216) | 0.799 (0.790) | 0.478 (0.380) |

but only becomes inferior to Gaussian OT in the nonlinear setting with at least 8 confounders. ConDo MMD disastrously explodes in all settings with at least 8 confounders.

We next augment the number of confounders by generating a noisy additive decomposition of our original confounder. We first uniformly sampling the "true" confounder as before, and generating the feature from it as before. We then generate a random multidimensional confounder summing to the "true" confounder of the desired dimensionality (Dickinson, 2010), and provide this to the ConDo methods. The results are shown in Figure 4. The results are very similar to those from the irrelevant confounder experiment. However,

Table 3: Summary of results for 1d data with 1d continuous confounder with noise

| | Noisy, Label-Shifted and Feature-Shifted | | |
| --- | --- | --- | --- |
| | Homoscedastic Linear | Heteroscedastic Linear | Nonlinear |
| Before Correction | 30.659 (23.746) | 31.474 (24.963) | 18.814 (13.150) |
| Oracle | 0.481 (0.441) | 0.487 (0.499) | 0.513 (0.483) |
| Gaussian OT | 9.459 (17.819) | 9.478 (11.663) | 5.694 (4.186) |
| MMD | 7.524 (5.539) | 9.063 (6.953) | 14.001 (9.732) |
| ConDo Linear-ReverseKL | 0.649 (1.224) | 1.608 (1.518) | 3.787 (2.517) |
| ConDo GP-ReverseKL | 0.697 (1.230) | 1.005 (1.365) | 2.262 (2.017) |
| ConDo PoGMM-ReverseKL | 1.036 (1.237) | 1.424 (1.391) | 2.381 (2.066) |
| ConDo MMD | 2.932 (2.136) | 4.925 (3.716) | 5.565 (3.609) |
| | Noisy, Label-Shifted Only | | |
| | Homoscedastic Linear | Heteroscedastic Linear | Nonlinear |
| Before Correction | 1.025 (1.036) | 1.006 (0.993) | 1.007 (1.025) |
| Oracle | 1.025 (1.036) | 1.006 (0.993) | 1.007 (1.025) |
| Gaussian OT | 9.546 (17.371) | 8.269 (11.043) | 4.621 (3.903) |
| MMD | 7.263 (5.682) | 8.512 (6.642) | 4.660 (3.715) |
| ConDo Linear-ReverseKL | 1.032 (1.056) | 1.340 (1.241) | 3.962 (3.100) |
| ConDo GP-ReverseKL | 1.032 (1.057) | 1.549 (1.352) | 1.737 (1.442) |
| ConDo PoGMM-ReverseKL | 1.282 (1.178) | 1.488 (1.341) | 1.294 (1.149) |
| ConDo MMD | 3.010 (2.509) | 3.265 (2.708) | 1.007 (1.025) |
| | Noisy, Feature-Shifted Only | | |
| | Homoscedastic Linear | Heteroscedastic Linear | Nonlinear |
| Before Correction | 23.855 (24.127) | 24.825 (24.400) | 15.086 (13.863) |
| Oracle | 0.509 (0.524) | 0.518 (0.477) | 0.473 (0.503) |
| Gaussian OT | 1.368 (1.407) | 1.787 (1.890) | 2.572 (2.323) |
| MMD | 2.671 (2.643) | 2.481 (2.339) | 8.949 (8.748) |
| ConDo Linear-ReverseKL | 1.252 (1.277) | 1.518 (1.558) | 2.880 (2.560) |
| ConDo GP-ReverseKL | 1.255 (1.275) | 1.468 (1.563) | 2.241 (2.096) |
| ConDo PoGMM-ReverseKL | 1.271 (1.292) | 1.507 (1.535) | 3.058 (2.726) |
| ConDo MMD | 1.357 (1.358) | 1.794 (1.756) | 4.100 (3.681) |
| | Noisy, No Shift | | |
| | Homoscedastic Linear | Heteroscedastic Linear | Nonlinear |
| Before Correction | 1.018 (0.954) | 0.970 (0.970) | 1.069 (0.978) |
| Oracle | 1.018 (0.954) | 0.970 (0.970) | 1.069 (0.978) |
| Gaussian OT | 1.403 (1.364) | 1.671 (1.705) | 2.215 (2.706) |
| MMD | 1.549 (1.538) | 1.249 (1.265) | 2.522 (2.783) |
| ConDo Linear-ReverseKL | 1.069 (1.016) | 1.192 (1.195) | 2.032 (2.435) |
| ConDo GP-ReverseKL | 1.081 (1.026) | 1.060 (1.081) | 1.370 (1.426) |
| ConDo PoGMM-ReverseKL | 1.110 (1.072) | 1.136 (1.125) | 1.320 (1.371) |
| ConDo MMD | 1.075 (1.046) | 1.023 (1.046) | 1.064 (0.978) |

in the nonlinear setting, the ConDo methods now start performing worse than Gaussian OT with only 4 confounders.

### 4.1.3   1d Data with 1d Categorical Confounder

Here, we generate 1d features based on the value of a 1d categorical confounder. We also use this setting to analyze the performance of ConDo for a variety of sample sizes. For each sample size under consideration, we run 10 random simulations, and report the rMSE compared to the latent source domain values (before applying the target-to-source batch effect).

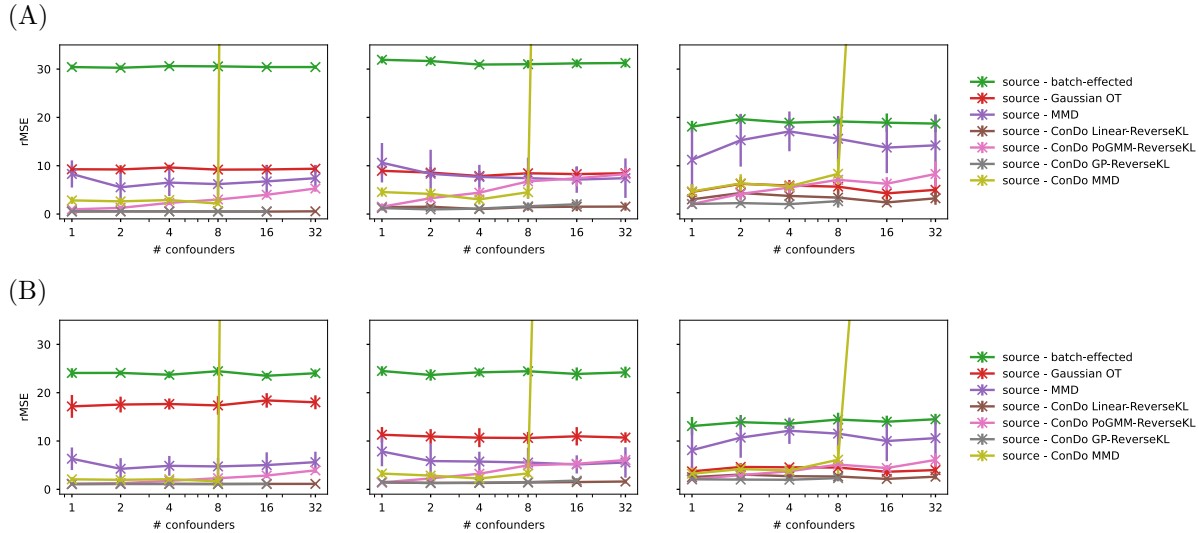

Figure 3: Results for transforming 1d data with multiple continuous confounders, with extra irrelevant $\mathcal{N}(0,1)$ confounders. Average rMSEs across 10 random simulations are shown for both training data (A) and heldout test data (B). The columns, in order, correspond to a confounder with a linear homoscedastic effect, a confounder with a linear heteroscedastic effect, and a confounder with a nonlinear heteroscedastic effect.

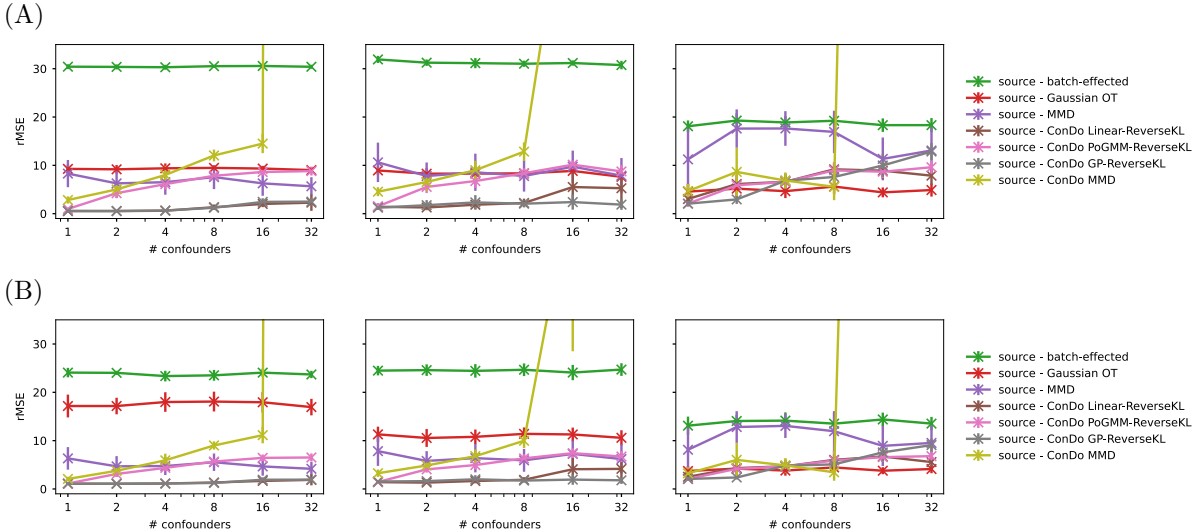

Figure 4: Results for transforming 1d data with multiple continuous confounders, with noisy additive decomposition. Average rMSEs across 10 random simulations are shown for both training data (A) and heldout test data (B). The columns, in order, correspond to a confounder with a linear homoscedastic effect, a confounder with a linear heteroscedastic effect, and a confounder with a nonlinear heteroscedastic effect.

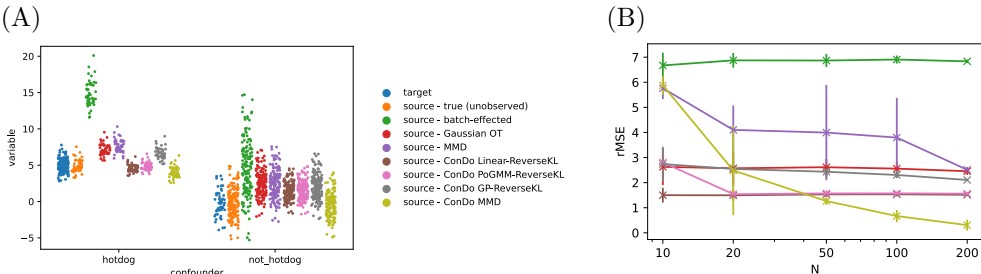

Figure 5: Results for transforming 1d data with a 1d categorical confounder. (A) Depiction of original, batch-effected, and domain-adapted data, for each value of the categorical confounder. (B) Plot of rMSE vs sample size for each of the domain adaptation methods. Each rMSE was averaged over 10 simulations, with the vertical lines indicating 1 standard deviation over the simulations.

Results are shown in Figure 5. In Figure 5(A) we see that with a 200 source (and 200 target) samples, the ConDo methods have converged on the correct transformation, while their confounding-unaware analogues do not. We see in Figure 5(B) that with even 10 samples, our ConDo Linear Reverse-KL method correctly aligns the datasets. Meanwhile, with at least 100 samples, all our ConDo methods have smaller rMSE. Overall, we see that the non-MMD ConDo methods are robust to small sample sizes.

### 4.1.4 Affine Transform for 2d Data with 1d Categorical Confounder

Next, we analyze the performance of our approach on 2d features requiring an affine (rather than location-scale) transformation. We also use this setting to assess the downstream performance of classifiers which are fed the adapted source-to-target features. The synthetic 2d features, before the batch effect, form a slanted "8" shape, shown in blue/green in Figure 6. Two linear classifiers, up-vs-down (in magenta) and left-vs-right (in cyan) are applied to this target domain.

Our results are shown in Figure 6. On the left column, we compare methods in the case where there is no confounded shift. (This setting is from Python Optimal Transport (Flamary *et al.*, 2021).) In the middle column, we have induced a confounded shift: One-fourth of the source domain samples come from the upper loop of the "8", while half of the target domain samples come from the upper loop. This allows us to assess the affects of confounded shift on downstream prediction of the confounder (up-vs-down), as well as a non-confounder (left-vs-right). In the right column, we have induced a confounded shift as before, while making the true source-target transform more challenging, by having a non-negative element in the transformation matrix.

We see that ConDo Linear-ReverseKL is the only method that has small rMSE and high accuracy in all settings. All methods perform similarly where there is no confounded shift, but the vanilla domain adaptation approaches fail in the presence of confounding.

## 4.2 Real Data

We compare ConDo to baseline methods on image color adaptation and on gene expression batch correction.

### 4.2.1 Image Color Adaptation

We here apply domain adaptation to the problem of image color adaptation, depicted in Figure 7. We start by adapting back and forth between two ocean pictures taken during the daytime and sunset (from the Python Optimal Transport library (Flamary *et al.*, 2021) Gaussian OT example). In this scenario, there is no confounding, since the images contains water and sky in equal proportions. Thus, conditioning on each pixel label (categorical, either "water" or "sky"), makes no difference, as expected. Next, we attempted color adaptation between the ocean daytime photo and another sunset photo including beach, water, and sky.

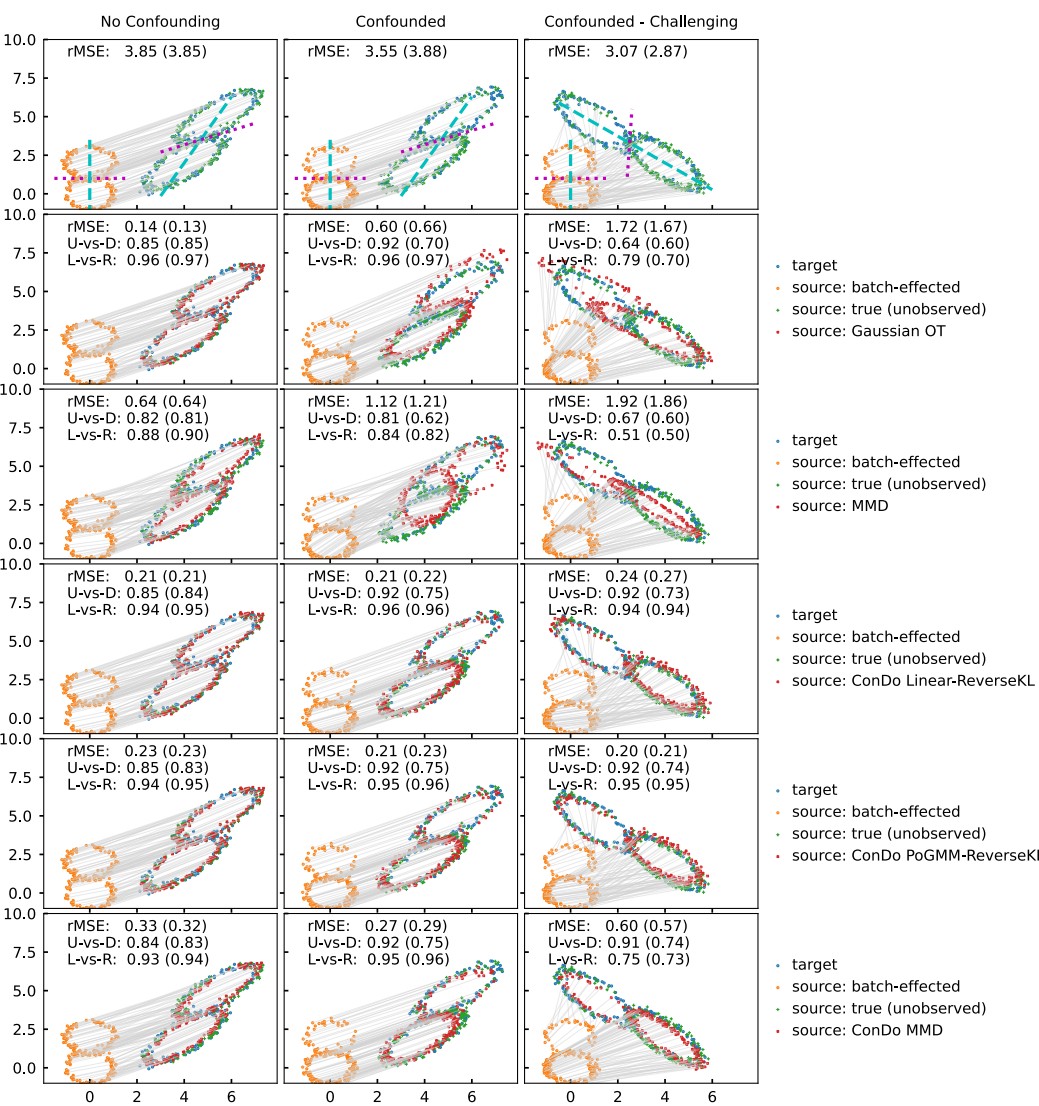

Figure 6: Results of affine transform of 2d data with a categorical confounder. We print the rMSE as well as the up-vs-down and left-vs-right accuracies on both the training data, and on heldout test data in parentheses. These values are the result of averaging over 5 random simulations, while the plot is generated from the final simulation.

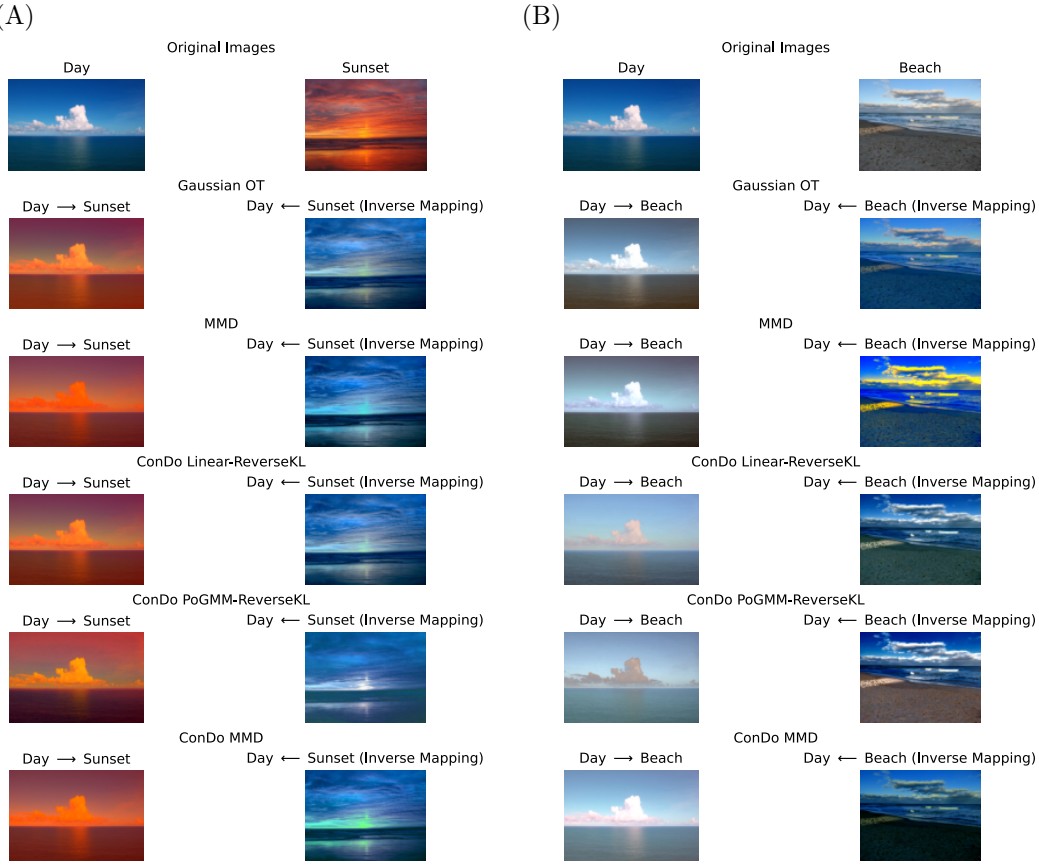

Figure 7: Image color adaptation results without (A) and with (B) confounded shift. The inverse mapping shown on the right columns are derived by inverting the already-learned mapping, not from learning a new mapping. We see that ConDo is non-inferior in (A). In (B), we see that non-ConDo methods produce gray-ish sky and white clouds in (Day → Beach) images and yellow clouds in (Day ← Beach) images. Meanwhile, reverse-KL ConDo methods produces light blue sky and peach/gray clouds in (Day → Beach) images and white clouds in (Day ← Beach) images.

Here, there is confounded shift, so ConDo successfully utilizes pixels labeled as "sky", "water", or "sand". More results, including a depiction of pixel labelling, are in Appendix D.5.

### 4.2.2  Gene Expression Batch Effect Correction

We analyze performance on the *bladderbatch* gene expression dataset commonly used to benchmark batch correction methods (Dyrskjøt *et al.*, 2004; Leek, 2016). We use all 22,283 gene expressions (i.e. features) from this bladder tissue Affymetrix microarray dataset; *bladderbatch* was preprocessed so that each feature is approximately Gaussian. In our experiment, we attempt a location-scale transform, as is typical with gene expression batch effect correction. We choose the second largest batch (batch 2, with 4 cancer samples out of 18 total) as the source domain, and the largest batch (batch 5, with 5 cancer samples out of 19 total) as the target domain. The confounder is 1d categorical (cancer or non-cancer).

Because the cancer fractions are roughly the same for batches 2 and 5, we do not expect to need to account for confounding. Results are shown in Figure 8(A). For each method, we visualize the effects of correction with t-SNE (Van der Maaten & Hinton, 2008) and PCA. We see that all methods are roughly equally successful at mixing together the samples from different batches (i.e., by color), while keeping cancer vs not-cancer samples clustered apart (i.e., X versus O). For each method, we also compute the silhouette scores of the

adapted datasets, with respect to the batch variable (and, in parentheses, the test result variable). We desire the silhouette score to be small for the batch variable, and big for the test result variable.

We repeat the experiment after removing half (7) of the non-cancer samples in batch 2, so that batch 2 is 4/11 non-cancerous, while batch 5 remains 5/19 non-cancerous. Results are shown in Figure 8(B). We see that ConDo linear Gaussian method performs better than vanilla Gaussian OT, and ConDo MMD performs better than vanilla MMD.

## 5 Related Work

As far as we are aware, previous work on domain adaptation does not address our exact problem. There is a large body of research in domain adaptation which maps both source and target distributions to a new latent representation where they match (Baktashmotlagh *et al.*, 2013; Yan *et al.*, 2017; Ganin *et al.*, 2016; Gong *et al.*, 2016). These however cannot achieve data backwards-compatibility, because they create a new latent domain. Other domain adaptation methods are also inapplicable to our setting since they match distributions via reweighting samples (Cortes & Mohri, 2011; Tachet des Combes *et al.*, 2020) or dropping features (Kouw *et al.*, 2016).

Prior research exists for performing domain adaptation when both features and label are shifted, including the generalized label shift (GLS) / generalized target shift (GeTarS) (Zhang *et al.*, 2013; Rakotomamonjy *et al.*, 2020; Tachet des Combes *et al.*, 2020). However, these methods assume the specific prediction setting where the label is the confounder, and optimize composite objectives that combine distribution matching and prediction accuracy. In our case, the confounder may not be the label of our prediction model of interest, and indeed we may not even be mapping covariates for the purpose of any downstream prediction task. Furthermore, by conditioning on confounders, our framework can handle multivariate confounders or even complex objects which are accessed only via kernels. Landeiro *et al.* introduced the term *confounding shift* to describe a form of GLS/GeTarS, but it does not match our *confounded shift* assumption, since the confounding variables are unobserved. Their method, which comprises confounder detection and adversarial confounder-robust classification, is substantially different from our approach.

The most appropriate domain adaptation methods for our context perform asymmetric feature transformation, in which source features are adapted to target features, and are thus compatible with general-purpose backwards compatibility. EasyAdapt (Daumé III, 2007) and EasyAdapt++ (Daumé III *et al.*, 2010) are notably successful approaches for such adaptations, but they employ concatenation, which presents difficulty in our setting. We expect to not have confounders available at inference time, which means that we cannot include them in the concatenated features.

Previous work which explicitly matches conditional distributions (Long *et al.*, 2013) instead uses the conditional distribution of the label given the features, rather than our approach of matching the features conditioned on the labels. It also constructs a new latent space, rather than mapping from source to target for backwards compatibility.

Our work is aligned in spirit with optimal transport with subset correspondence (OT-SI), which implicitly conditions on a categorical confounder (the sample's subset) to learn an optimal transport map (Liu *et al.*, 2019b). Our approach explicitly conditions on confounders and is more general, allowing continuous, multivariate, and (using kernels with our GP and MMD based methods) even general objects as confounding variables.

## 6 Conclusion and Future Work

### 6.1 Conclusion

We have shown that minimizing expected divergences / distances after conditioning on confounders is a promising avenue for domain adaptation in the presence of confounded shift. Our proposed use of the reverse KL-divergence and our dynamic choice of RBF kernel bandwidth are (to our knowledge) new in the field of domain adaptation, and may be more broadly useful. Focusing on settings where the effect

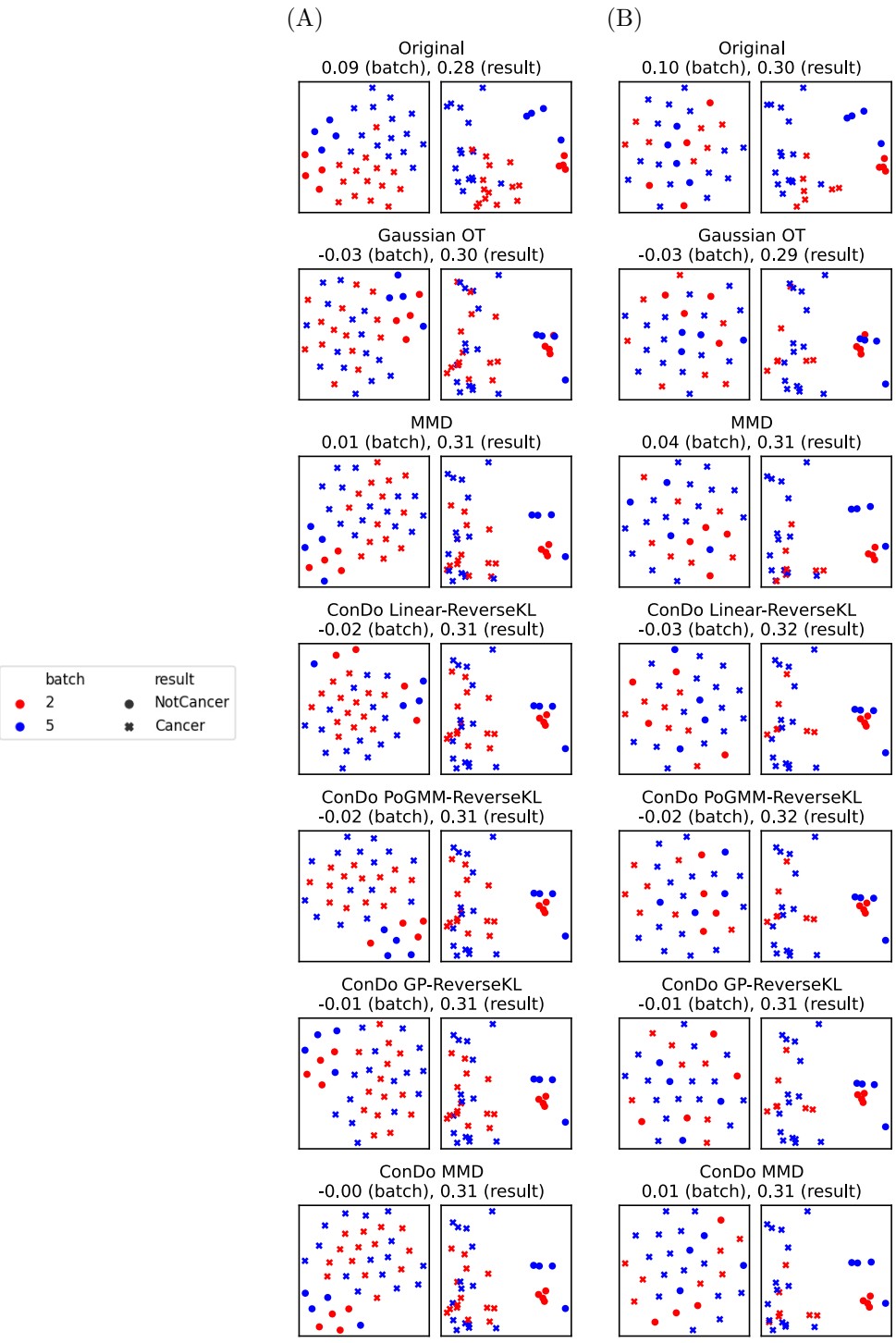

Figure 8: Results on *bladderbatch* dataset, without confounded shift (A), and with confounded shift (B). We would like reds and blues to be well-mixed, while cancer and non-cancer samples to cluster apart. In both (A) and (B), we show t-SNE on the left, and PCA on the right.

of the confounder is possibly complex, yet where the source-target domains can be linearly adapted, we demonstrated the usefulness of both parametric and nonparametric algorithms based on our framework. Our ConDo framework seems to learn adaptations that are good for a variety of downstream tasks, including prediction and clustering.

## 6.2 Future Work

Due to the affine restriction on the transformation, our proposed approach is more appropriate for adaptation settings where source and target correspond to different versions of sensor devices, different laboratory protocols, and similar settings where the required adaptation is affine (or even location-scale). It would be useful to examine whether our framework extends gracefully to nonlinear adaptations, such as those parameterized by neural networks.

Both our proposed divergences, the reverse KL-divergence and the MMD, suffer from non-identifiability. Multiple transformations may match the source distribution to the target distribution. Both our objective functions are indifferent among such transforms, and we currently rely on gradient flow from the initial parameters to make a sensible choice. OT offers a principled criteria, minimal transport cost, to choose among transformations which provide equal fit to the data. The squared 2-Wasserstein distance is an especially promising alternative distance function. Minimal transport cost is an excellent "prior" but not the only defensible choice. $L_p$ regularization, empirical Bayes weight sharing such as used by ComBat (Johnson et al., 2007), and constraints (e.g. non-negativity or zero-off-diagonal) may instead be preferred, and may be fruitfully combined with KL-divergence and/or MMD.

Our methods based on the KL-divergence and MMD could also admit further improvement. Our KL-divergence method relies on either a (potentially multivariate) linear Gaussian distribution or a univariate (nonlinear) Gaussian Process. Extending the latter to full affine transformations of multivariate features could take advantage of recent advances in using Gaussian Process conditional density estimation (Dutordoir et al., 2018) for better modeling of uncertainty, and recent advances in improving scalability for multivariate outputs (Zhe et al., 2019). Optimization of MMD is challenging, because it is a nonconvex functional. Our sampling from the confounder prior injects noise which may help overcome the nonconvexity, but adding Gaussian noise to the samples has been proven to be beneficial (Arbel et al., 2019), so it is worth examining.

We believe that our framework can be extended for a wide array of applications. For example, Wasserstein Procrustes analysis was recently developed and applied to align text embeddings across languages (Grave et al., 2019; Ramírez et al., 2020). By combining this with conditioning on confounding variables, one could potentially align embeddings between languages with different topic compositions. In addition, our approach could potentially be applied to one of the MMD's most prominent applications – two-sample testing (Gretton et al., 2012) – in the presence of confounding.

Finally, thus far our analysis of ConDo has been purely empirical. Theoretical analysis would surely be appropriate, particularly before applying it to data analyses and statistical inference tasks.

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
