# OpenReview forum: "Towards Backwards-Compatible Data with Confounded Domain Adaptation"
_TMLR — Rejected by TMLR_

### Review · Reviewer_X3DL · 2022-04-26

**Summary Of Contributions:**

# Summary
This paper studies a type of domain adaptation problem, called confounded shift, in which the conditional distribution $p(x|y)$ of the covariate $X$ given another variable (confounder) $Y$ in the target domain equals that of the covariate transformed by some unknown function $g$.
The marginal distributions of the covariates or the confounder can change from the source domain and the target domain.
The authors propose a method for finding the transform $g$ using $(X, Y)$-data collected from both source and target domains.
The proposed method tries to find $g$ that minimizes the reverse KL-divergence or the Maximum Mean Discrepancy (MMD) between the conditional distribution of the two domains.
The authors confirm that the proposed method can accurately learn the transform in each of the settings with one- and two-dimensional synthetic data with a one-dimensional continuous and categorical confounder.
The authors also apply the proposed method to two real-world datasets and demonstrate its usefulness.


**Broader Impact Concerns:**

I do not find any ethical concerns with this submission.

**Requested Changes:**

# Requested changes
## Major points
- Please explain the motivation for choosing KL-divergence and MMD over optimal transport.
- Please explain why the proposed methods do not suffer from the identifiability problem although they only try to match the distributions. For example, horizontally flipping the source data points first and then applying the transform presented in Figure 3 would give equally good divergence scores in the 8-like shape example. What makes the methods choose the non-flipped one?
- Please check if Eqs. (13), (14), and the corresponding code are correct (regarding my concern about the log-determinant term).
- Page 8, "To handle multivariate categorical $Y$, we take the product of each conditional distribution": I could not understand this part well. Does this mean that the method makes a sort of conditional independence assumption? Please make the description clearer perhaps using equations. In particular, if there is any assumption made by the method, it should be clearly explained.

## Minor points
- The proposed method assumes the transform is affine, which may be restrictive.
- In Eqs. (7-10), is it $\delta(y - Y_S^{(n)})$ and $\delta(y - Y_T^{(n)})$?
- Page 6, "The product prior requires smoothing because the empirical distributions may have non-intersecting support": I could not understand this. Please give more details.
- In the caption of Figure 4, "We see that Condo Reverse-KL methods give the best results": How do the authors conclude that?

**Strengths And Weaknesses:**

# Strengths:
- The problem setup is interesting and seems practically useful.
- The paper provides a series of interesting experiments whose results supporting the superiority of the proposed method over the existing methods, Gaussian OT and MMD, which do not take into account the confounded shift.

# Weaknesses:
- The motivation for choosing KL divergence and MMD over optimal transport is not clear. Minimizing the KL divergence or MMD cannot generally identify the transform because there can be many transform that result in the same distribution. On the other hand, optimal transport can break such ties by looking at the transport cost. Without identification, the classifier trained in the source domain and adapted to the target domain may not work.
- I also have a concern with the correctness of the KL-divergence-based method:
I am not convinced that Eqs. (13) and (14) are totally correct. The forward- and reverse-KL divergences must be defined for any non-singular matrix A, even with a negative determinant. However, the log-determinant terms of Eqs. (13) and (14) are not defined for A with a negative determinant. I think a positive semi-definite matrix should appear inside the log-determinant.
- The proposed method assumes the transform is affine, which may be restrictive.
- For some of the experiments, it is hard to see the improvements that the authors claim.

---

> ### Author Response · Authors · 2022-05-05
> **Reply to Review by X3DL**
>
> Our shared reply will address concerns related to KLD/MMD vs OT, and the restriction to affine adaptation.
>
> - RE "the identifiability problem ... What makes the methods choose the non-flipped one?"
>
> See also our shared response.
>
> Concretely, the methods chose the non-flipped one in our experiments, not due to the optimization objective alone. Rather it was due to the gradient flow induced by the objective combined with our initialization, the identity transform. (And in the case of location-scale reverse-KL, which is closed-form not optimized, we explicitly choose the positive scale coefficient.)
>
>  - RE log-det term in Eqs. (13), (14):
> Thanks for catching this. Mathematically, the implication of $log(det(A)^2) = 2 log(|det(A)|) \ne 2 log(det(A))$ is that both forward-KL and reverse-KL are defined for A with negative-determinant. Computationally, since we always initialize A with the positive-determinant identity matrix, it remains true that reverse-KL maintains positive-determinant A across optimization iterations with the log-barrier, while forward-KL tends to produce iterates where the sign of the determinant oscillates. We have updated Section 3.4.1 to incorporate this fix.
>
>
> - RE Page 8 (Product of Gaussian Mixture Models):
> We have added a more precise explanation with explicit assumption / equation in Section 3.5.2.
>
>
> - RE Dirac delta notation in Eqs. (7-10):
> Thanks for catching this; we have fixed it.
>
>
> - RE Page 6 ("product prior requires smoothing"):
> We have added the following motivation for prior smoothing to Section 3.3: "For the product prior, we note that the empirical distributions may have non-intersecting support. Without smoothing the priors, this would be problematic because (for example) the weight $\mathbf{w}^{(n)}_S$ attached to a particular sample $Y^{(n)}_S$ would be zero unless $Y^{(n)}_S = Y^{(i)}_T$ for some $1 \le i \le N_T$. This happens almost never if $Y$ comes from a continuous distribution. Therefore, before computing their product, we smooth the priors using the confounder-space kernel $k\_{\mathcal{Y}}$ as follows:"
>
> Please let us know if would like additional explanation beyond the above.
>
> You might be wondering whether this would only work if the true distribution of Y is the product of the source and target domains. The answer is no, due to the Confounded Shift assumption
> ($\forall y \in \mathcal{Y}, \mathcal{D}_S(g(X)|Y=y) = \mathcal{D}_T(X|Y=y)$). The chosen prior distribution does not need to match the true distribution over $Y$ for the true affine transformation $g(X)$ to be a minimizer of the objective. To see this, suppose that the estimated conditional distributions equal the true conditional distributions ($\hat{\mathcal{D}}_S = \mathcal{D}_S, \hat{\mathcal{D}}_T = \mathcal{D}_T$). Then even if we choose the prior to equal some singular element $y \in \mathcal{Y}$, the KLD would be minimized to 0 by the true $g(X)$. So we can restrict the prior to any subset of $\mathcal{Y}$, but we would expect to pay a price in terms of sample complexity.
>
> We might also add the above paragraph to Section 3.3.
>
>
> - RE Figure 4 (image color adaptation -- now Figure 6 in revised manuscript):
> We have updated the caption to the following: "We see that ConDo is non-inferior in (A). In (B), we see that non-ConDo methods produce gray-ish sky and white clouds in (Day $\rightarrow$ Beach) images and yellow clouds in  (Day $\leftarrow$ Beach) images. Meanwhile, reverse-KL ConDo methods produces light blue sky and peach/gray clouds in (Day $\rightarrow$ Beach) images and white clouds in (Day $\leftarrow$ Beach) images."

---

> > ### Comment · Reviewer_X3DL · 2022-05-22
> > **The authors have mostly addressed my concerns**
> >
> > I have read the authors' response to the concerns that I raised in my initial review comments. The authors' replies have addressed my major concerns. I have two comments.
> >
> > 1. This is a small suggestion. About the choice of the discrepancy, besides the points that the authors have written in the paper, I think the following argument is worth adding to the paper as well:
> > > when non-identifiability leads to multiple solutions which provide equal fit to the data, minimal transport cost is an excellent "prior" but not the only defensible choice. L_p regularization, empirical Bayes weight sharing (used by ComBat), constraints (eg non-negativity or zero-off-diagonal) may instead be preferred, and may be fruitfully combined with KLD and/or MMD
> >
> > 2. About the identifiability issue in the experiments, the authors claim,
> > > Rather it was due to the gradient flow induced by the objective combined with our initialization, the identity transform. (And in the case of location-scale reverse-KL, which is closed-form not optimized, we explicitly choose the positive scale coefficient.)
> >
> > This sounds reasonable, but it raises another question. Are the proposed optimization problems convex? If they are, I don't think there can be multiple isolated local optima as the authors describe above.
> > On the other hand, the authors criticize MMD as "because MMD is a non-convex functional, it tends get stuck in local minima," which could be interpreted as saying other approaches lead to convex optimization.

---

> > > ### Author Response · Authors · 2022-06-01
> > > **Reply to Comment by X3DL**
> > >
> > > 1. Thanks for this suggestion. We have revised the Future Work section.
> > >
> > > 2. Neither the proposed optimization problems are convex, as we have verified by plotting the objective in the 1d case. Anecdotally, only reverse-KL seems to be benignly non-convex.

---

### Review · Reviewer_55pu · 2022-05-03

**Summary Of Contributions:**

This paper proposes a novel assumption for dataset shift adaptation called confounded shift, which is the situation where the class-conditional probability of the source and target domains are identical if we can find an input mapping g and applies it to the source data before evaluating the class-conditional probability, i.e., $\mathcal{D}_{S}(g(X)|Y=y)=\mathcal{D}_T(X|Y=y)$.

To solve this problem, this paper proposes a framework to effectively find a good prediction function (e.g., classifier, regressor) by mapping the data in the target domain to be more similar to the source domain, and then use the best hypothesis in the source data to perform the prediction.

The main idea to find the mapping is to assume that the mapping is affine (Ax+b), where A is a diagonal matrix (location-scale adaptation). Then, the method is to minimize the divergence between the class-conditional probability of the source and target domains based on equation 6. Equation 6 has flexibility over (1) the choice of the prior probability distribution, (2) the choice of divergence, (3) the choice of how to model the class-conditional probability.  Several choices were implemented and evaluated empirically in the experiment section.

**Broader Impact Concerns:**

This paper is a technical work that makes assumptions about the confounded shifts. I am not sure if there is an important broader impact concern to be discussed. If I haven't missed it, this paper in the submission form also does not include such a discussion. Perhaps one may think about harm that may cause when we assume this assumption if there is any.

**Requested Changes:**

1. Problem setting should be clearly stated. In this problem, what are input, output, constraint, objective. For example,
- Do we have target labels for all target data or just a subset of them?
- What is the expectation of a number of data in the target domain (much much smaller than the source domain?). If target dataset size is sufficient to learn a good hypothesis, then learning it from scratch could be more effective sometimes since using source data can incur negative transfer.
- What is the objective, e.g., find the mapping g that minimizes the expected (regression/classification) risk with respect to target domain by using the composition g o h, where h is the best and fixed hypothesis in the source. Clear statements would make the problem setting clear and highly useful to the readers.

I am aware that it could be written in the paper but I think clear mathematical problem formulation is not explicitly stated.

2. It would be important to note that covariate shift adaptation expects to have only unlabeled data in the target domain (although the dataset size can be large). Thus, covariate shift adaptation may not always be easier than the confounded shift setting here.

3. I think it is incorrect to say that the proposed confounded shift is a special case of generalized label shift (GLS). From Table 1 (I really appreciate that Table 1 is given since it is clear and easy to understand the difference between assumptions), we cannot find g(X) that easily recovers Confounded shift assumption unless we give domain label into g as well. Thus, dropping the sentence that this setting is a special case of GLS is better.

4. Is there any experiment that justifies the usefulness of using a dynamic strategy for choosing kernel bandwidth? Changin bandwidth means changing objective function to be minimized, which might make the target optimization problem more difficult and confusing.

5. In 3.4.1, the reason to choose KL-divergence is not very convincing to me. It is written something likes KL-div is zero if the adaptation is successful. But isn't that true for any reasonable divergence? I think it is better to argue about other things that other divergences may not be able to give us, e.g., computational efficiency.

6. Some parts refer to Appendix without given the section of the appendix, e.g., the end of section 4.1.1, 4.2.1

7. Typos: figure 2: ategorical -> categorical,

**Strengths And Weaknesses:**

Strengths
1. I think the shift assumption provided in this paper is novel. Although I have some concerns about practicality because it makes me feel the target domain information should be at most as informative as the source domain because one can find the mapping in the source domain to match the class-conditional probability of both domains.
2. Proposed framework is intuitively understandable and is reasonable to be effective if the assumption holds. Without theoretical guarantee, I am convinced that the method should be useful.
3. I appreciate the fact that several choices of configurations for the framework were explored (see Section 3.3, 3.4, 3.5) and evaluated in section 4.

Weaknesses
1. No theoretical evaluation of the proposed framework. But I think it is still acceptable to accept a paper that is purely empirical if it has sufficiently convincing experiments.

2. Dataset used in this experiment looks a bit weak in my opinion. Synthetic data has one dimension or two dimensions and the labels have only one dimensional for the synthetic data. This makes me wonder if the proposed method can be ineffective when the data has high dimension. And when we have only one label, I think it is possible to compare the method with methods for label-shift adaptation, or covariate shift adaptation. For real datasets, I found that dataset information does not include, e.g., data dimension. I am aware that bladderbatch has been used, but it would make the paper more self-contained to describe the dataset clearly.

3. Comparisons with other methods can be improved. In my understanding, currently only the proposed method utilize the information of
given labels as MMD and Gaussian OT did not use it. Please correct me if I am wrong.

4. There is no experiment to validate the performance of a classifier. We only compare the input space. I was wondering if one directly trains a classifier using only target domain data versus this proposed adaptation, will they perform similarly in the experiment of this paper?

5. Literature review of dataset shift could have been more extensive. I think there are many more assumptions on dataset shift [1][2]

[1] Quiñonero-Candela, J., Sugiyama, M., Schwaighofer, A., & Lawrence, N. D. (Eds.). (2008). Dataset shift in machine learning. Mit Press.

[2] Kull, M., & Flach, P. (2014). Patterns of dataset shift. In First International Workshop on Learning over Multiple Contexts (LMCE) at ECML-PKDD.
I think section 2.4 could be more extensive. In its current form, it's not very informative. We could discuss more about the assumption of input/output for each problem, e.g., covariate shift adaptation usually assumes that we can have a lot of target data but with zero labels in it, etc.

6. I have a doubt if the assumption of confounded shift is too strong. I understood the importance of backward compatibility, but from the assumption, does that mean the target domain has information strictly less than the source domain when thinking from the perspective of class-conditional probability? If so, then why one would update the sensor to have weaker information than the previous one?

7. Assumption on g is very strong (location-scale adaptation), I feel this is a big disadvantage of the proposed approach (but nothing wrong with the proposed confounded shift assumption). For the first work in this line, I still think it is acceptable to first focus on location-scale adaptation.

---

> ### Author Response · Authors · 2022-05-05
> **Reply to Review by 55pu**
>
> - RE "Problem setting should be clearly stated."
>
> We agree, and have added answers to these questions in the revised Section 3.2 (Main Idea).
>
> - RE "Dataset used in this experiment looks a bit weak in my opinion. Synthetic data has one dimension or two dimensions and the labels have only one dimensional for the synthetic data.":
>
> We have added (Section 4.1.2) a set of synthetic data experiments where the confounders vary in dimension. We find that our simple conditional estimators (eg Linear ReverseKL) generally scale well with confounder dimension.
>
> - RE "For real datasets"
>
> We have added dimensionality / preprocessing info on image color adaptation and bladderbatch to the main text and appendix.
>
> - RE "There is no experiment to validate the performance of a classifier."
>
> Section 4.1.4 (Affine Transform for 2d Data) contains two such experiments. We evaluate classification performance on a label that is provided to ConDo (up-vs-down) and on a label that is not provided (left-vs-right). Section 4.2.2 also contains such an experiment, since silhouette scores are related to accuracy based on nearest-neighbor classification. The "result" (cancer vs non-cancer) silhouette score in Figure 7B (with confounded shift) is highest for ConDo Linear-ReverseKL and ConDo PoGMM-ReverseKL.
>
> - RE "I was wondering if one directly trains a classifier using only target domain data versus this proposed adaptation, will they perform similarly in the experiment of this paper?"
>
> Our left-vs-right experiment exemplifies the setting where the V2 data is unlabelled, so it would be impossible to train on only V2 domain data. And it also exemplifies a setting where training using only V1 data (which is labelled) would fail catastrophically on V2 data.
>
> - RE "Literature review of dataset shift"
>
> Thank you for pointing this out. We have updated Section 3.1 (Our Assumption) to describe our setting as a combination of prior probability shift and covariate observation shift as defined in
> (Kull & Flach, 2014).
>
> - RE "I understood the importance of backward compatibility, but from the assumption, does that mean the target domain has information strictly less than the source domain when thinking from the perspective of class-conditional probability? If so, then why one would update the sensor to have weaker information than the previous one?"
>
> Yes, we expect that the target domain has strictly less information than the source. But no, we assume the updated sensor has stronger information. This is subtle, because we follow common nomenclature and adapt the "source" to the "target." So the "source" is the updated V2 sensor with stronger information (but fewer samples), while the "target" is the old V1 sensor. Typically in DA the target has few (or no) labeled samples. But in our setting, the source has few labeled samples (or samples annotated with only with the confounder, not the prediction label).
>
> - RE "Assumption on g is very strong (location-scale adaptation)"
>
> We do not always restrict g to be location-scale. For example, Section 4.1.4 (Affine Transform for 2d Data) and Section 4.2.1 (Image Color Adaptation) involve full affine adaptation.
>
> - RE "we cannot find g(X) that easily recovers Confounded shift assumption unless we give domain label into g "
>
> We cannot envision a situation in which one does not know whether a sample comes from the source or target domain, but you are correct so we have changed the text. Thanks for catching this.
>
> - RE "experiment that justifies the usefulness of using a dynamic strategy"
>
> We have added this in Appendix D.1, and referred to it from the main text.
>
> - RE "the reason to choose KL-divergence"
>
> See our shared response.
>
> - RE "the section of the appendix" and "ategorical -> categorical"
> Thanks for catching -- fixed.

---

> > ### Comment · Reviewer_55pu · 2022-05-30
> > **Thank you for the clarification and addressing my concerns**
> >
> > The authors have modified the draft to (1) clarify the problem setting (in 3.2 + figure 1), (2) clarify my misunderstandings for the experiment part (4.1.4) and the assumption on g, and (3) slightly added the paper to discuss the relationship between the proposed assumption and the existing assumptions in domain adaptation (in 3.1).
> >
> > I feel the draft has been improved and I can appreciate the contribution more. I found that at first, I had a hard time thinking in a way that V1 is the target (with large data size) and V2 is the source (with small data size but the sensor has more information), where the goal is to improve the performance w.r.t. target distribution V1. Because it is more common to think that the target data is insufficient to learn a good hypothesis while the source data is sufficient to do so w.r.t. error in the source distribution. However, the notion in this paper is used in quite an opposite way (I still feel that using the terminology this way is a bit confusing).

---

> > > ### Author Response · Authors · 2022-06-01
> > > **Reply to Comment by 55pu**
> > >
> > > Unfortunately, the terminology is confusing whether we choose (V1=target, V2=source) as in the manuscript, or (V1=source, V2=target). The problem with the latter is that we would then be transporting the target to the source, which is also confusing, and especially so in the context of transport theory.

---

### Review · Reviewer_tFRV · 2022-05-04

**Summary Of Contributions:**

This paper proposes a new domain adaptation setting (DA) called confounded shift. The idea of this setting is to achieve several goals: 1) have an adaptor for two domains that doesn't depend on the confounder (label) to make it useful for a variety of downstream tasks; 2) address generalised target shift (GLS) setting where the adaptor should match conditional target distribution given cofounder; this is more useful than the GLS setting of (Tachet de Combes et al. 2020) as it allows to augment target data with source-aligned data and use it eventually for other downstream tasks. The proposed idea is implemented algorithmically using the maximum mean discrepancy (MMD) and reversed KL divergence minimization between conditional distributions of the two domains given some prior on the confounder and by modeling the adaptor as an affine transformation. The results on several toy examples and the task of color transfer highlight the usefulness of the proposed approach.

**Requested Changes:**

Overall, I like the main idea of this paper but I think that it requires a major revision to better present the proposed ideas and relate them to the state-of-the-art. I would like to encourage the authors to do so based on the remarks outlined below.

I. Improving the motivation part of the paper and better relating it to existing literature: CRITICAL

In the introduction, the authors make the following 4 claims motivating their work:
1. “for tasks that do not involve predicting the confounder, we cannot simply perform standard covariate-shift domain adaptation, because the source and target datasets should not look alike.”
There exist methods, that align two domains without depending on the labels in either domain or learning a mapping dependent on the classifier. In such cases, the labels are then used only to learn a classifier afterward. Some common examples: Subspace Alignment, Optimal Transport for Domain Adaptation (without LpL1 regularisation).  Also, when the confounder for the new task is not observed, domain adaptation can fail but that’s out of reach for the unsupervised algorithms anyway.  This particular obstacle seems to be handled in this work by minimizing over a prior distribution of confounders rather than available information about the confounders but the prior seems to depend on Ys and Yt only.  It would be great to elaborate on this.

2. “for these other tasks that do not involve predicting the confounder (e.g. seizure risk), we cannot assume that the confounder (seizure risk) is known for all samples on which we will apply our adaptation.”
Why wouldn’t semi-supervised DA methods work in this case? It seems that once again this is related to the first point where we may need supervision for the shift between the confounder of the two tasks and where the prior seems to be defined for observed confounders only.

3. “we cannot discard information unrelated to predicting the confounder.”
This idea seems to suggest learning a push-forward from source to target while keeping the latter unchanged. This is known as an asymmetric approach in the DA literature (Weiss et al. 2016), and there are methods capable of proceeding this way, such as OTDA mentioned above that projects source to target via barycentric mapping while keeping the latter unchanged.

4. “we must make the V2 “backwards-compatible” with models trained on V1 data, producing a V2-to-V1 adapter that is then composed with V1-trained prediction models.”
Does this obstacle refer to the invertability of the aligning mapping? As before, I struggle to understand the motivation behind this point.

In general, my understanding is that the proposed setting is close to something that one could have termed “asymmetric generalized target shift” using the terminology of the existing literature as according to Table 1 the lack of symmetry with respect to g is the only difference here wrt Tachet des Combes et al. 2020.

II. Explaining why KL divergence works with affine transform for non-affine relationships between two domains: CRITICAL
I appreciate the idea of having a linear relationship between the two domains' input spaces when mapping their conditional distributions. My understanding is that for MMD this is compensated in part by projecting to a richer similarity-induced space where the relationship between the two domains may indeed be linear. But why is it supposed to work with KL divergence? I don’t fully understand how that can be the case (Toy examples with 8s, for instance, rightmost column) when the simulated data is linked through a non-linear transformation. Is there any explanation for this?

III. I think that there are several missing important references in this work that can be compared.
First, the term confounding shift was introduced in Landeiro et al. 2019 (Discovering and Controlling for Latent Confounds in Text Classification Using Adversarial Domain Adaptation, SDM) where a confounder was influencing both covariates and targets.
Second, the statement regarding the lack of work on minimizing MMD in DA with conditional distribution matching is not exactly true. I was surprised to see that Long et al. 2013 was not considered in this family as they factorize the joint distributions into two terms with one leading to marginal distributions discrepancy minimization while the other being the conditional distributions minimization using target pseudo-labels (Section 3.2.3).

Finally, I do not see why the authors didn’t try Zhang et al. 2013 method in this context (also their code is available online): it seems that their GeTarS model is perfect for their setting. Also, they mention that this work minimizes only the discrepancy between marginal distributions but eq. 12 from this paper clearly shows that they minimize both marginal and conditional distributions.
As such, TMLR doesn’t require to justify for the novelty of the proposed approach but I think that it is important to really highlight the distinct features of the proposal. In this case, it is hard to derive something conclusive about this as the authors limit their experiment's study to really naive baselines.

**Strengths And Weaknesses:**

Strengths:
- introduction of a new setting for generalized target shift
- interesting learning approach that aligns conditional distributions to solve GTS with respect to confounders drawn from a prior distribution
- nice experimental results in a variety of settings

Weaknesses:
- clarity of the manuscript can be improved
- some missing literature references

---

> ### Author Response · Authors · 2022-05-05
> **Reply to Review by tFRV**
>
> RE - "1... There exist methods, that align two domains without ..."
>
> It seems that there are two ways to use such methods in our setting. First, one could ignore the confounders when learning an alignment, which seems to waste information. Second, one could concatenate the confounder(s) to the features, and then learn an alignment. However, even for an affine transformation (on the concatenation), one would need to know the confounder values at test time, which is not possible in our setting.
>
> RE - "2... Why wouldn’t semi-supervised DA methods work in this case?"
>
> The semi-supervised DA setting is different from our setting in this case. In our setting, we have V1 dataset with features, confounders, and labels; a V2 training dataset with features and confounders; and a V2 testing dataset with features only. Thus, we have 3 distinct availability patterns among the three datasets. In semi-supervised DA, there are two availability patterns. One might have a V1 dataset with features and labels; a V2 training dataset with features and labels; and a V2 testing dataset with features. Thus, 2 availability patterns among the 3 datasets. Or one might have V1 dataset with features, confounders, and labels; a V2 training dataset with features, confounders, and labels; and a V2 testing dataset with features and confounders. Thus, again 2 availability patterns among the 3 datasets. (This is not to say that semi-supervised DA methods cannot somehow be applied; a future paper may very well show an approach inspired by EasyAdapt++ to be superior to ConDo in this case.)
>
> RE - "3... asymmetric"
>
> Thank you for bringing this to our attention. We added a discussion of asymmetric feature transformation methods to Related Work.
>
>
> RE - "4... “we must make the V2 “backwards-compatible” with models trained on V1 data, producing a V2-to-V1 adapter that is then composed with V1-trained prediction models.” Does this obstacle refer to the invertability of the aligning mapping? As before, I struggle to understand the motivation behind this point."
>
> Yes, this is one obstacle that precludes using UDA to transform the V1 to V2, then training a new V2-domain model on the (V1-to-V2 features, V1 labels) dataset. Another obstacle (common in certain industry settings), is that one is not able to immediately update the (V1-trained) model. One team is responsible for the part of the pipeline emitting features, but has no responsibility or control over the part of the pipeline emitting predictions from features. The V1-to-V2 update and the prediction model update are not synchronized, forcing interim backwards-compatibility.
>
>
> RE - "II. Explaining why KL divergence works with affine transform for non-affine relationships between two domains... Toy examples with 8s"
>
> In all our synthetic experiments, the true relationships between domains are affine. This is not immediately obvious in the Toy-8s experiment, because points are transported with intersecting paths. But parallelograms are in fact preserved by the affine rotation.
>
> RE - "Landeiro et al. 2019"
>
> Thanks for pointing this out. We have added it to Related Work.
>
> RE - "surprised to see that Long et al. 2013 was not considered in this family"
>
> We addressed this in Related Work: "Previous work which explicitly matches conditional distributions (Long et al., 2013) instead uses the conditional distribution of the label given the features, rather than our approach of matching the features conditioned on the labels. It also constructs a new latent space, rather than mapping from source to target for backwards compatibility."
>
>
> RE - "Also, they mention that [Zhang et al. 2013] minimizes only the discrepancy between marginal distributions but eq. 12 from this paper clearly shows that they minimize both marginal and conditional distributions."
>
> Perhaps we misunderstand Zhang et al. 2013, but we do not think eq. 12 minimizes difference between the source and target conditional distributions. The first term
>
> $$J=\|\mu[P^{new}\_X] - \mu[P^{te}\_X]\|^2$$
>
> which is a marginal distribution. The second term $J^{reg}$ is the regularization penalty in Eq 10 on the location-scale adaptation parameters. It minimizes the difference between $P^{new}\_{X|Y}$ and $P^{tr}\_{X|Y}$, not between $P^{te}\_{X|Y}$ and $P^{tr}\_{X|Y}$ as stated in the paper; we suspect it was a typo. In other words, they minimize the difference between the source and adapted-source conditionals, which is different from us.
>
> RE - "why the authors didn’t try Zhang et al. 2013"
>
> We had looked into this but did not for a few reasons: 1) We were unable to successfully run the authors' Matlab code on our system. 2) It is not compatible with full affine transformations. 3) Our experiments are focused on determining feasibility of matching conditionals vs marginals via Gaussianity (ie Gaussian ReverseKL vs Gaussian OT) and via kernels (ConDo MMD vs MMD). 4) As we state above, we do not believe Zhang et al matches source / target conditionals.

---

> > ### Comment · Reviewer_tFRV · 2022-05-23
> > **Thanks for clarification**
> >
> > I want to thank the authors for their clarification. I found Fig 1 very helpful in understanding the proposed framework.
> >
> > Before adding more comments, I would like to ask the authors to provide an updated version of the manuscript with highlighted changes. It would be extremely helpful to write the final review and assess what was done and what is left to be done.

---

> > > ### Author Response · Authors · 2022-05-24
> > > **Uploaded diff**
> > >
> > > We have updated the Supplementary Material to include a highlighted diff between the original submission and the revised manuscript.

---

### Author Response · Authors · 2022-05-05
**Shared reply to reviewers**

Firstly, we thank all three reviewers for their careful reading and helpful comments. We have incorporated their feedback in our revised manuscript.

Second, we agree that it would be interesting and likely beneficial to instead use the optimal transport, i.e. Wasserstein distance as the distance/divergence function. However, we chose to mention applying ConDo to Wasserstein Procrustes (Grave et al., 2019; Ramírez et al., 2020) as future work in our Conclusion for a few reasons. First, computational reasons. While KLD is especially problematic for not being geometry-aware, our ConDo location-scale KLD has closed-form solution so has comparable runtime to limma and ComBat, widely-used methods for batch correction in genomics. ConDo Wasserstein Procrustes presents computational challenges and opportunities in terms of optimization and computational efficiency; we think addressing these (presumably by interpolating MMD and OT with Sinkhorn) would merit separate treatment in a different paper. Second, in the fields where ConDo is likely to be most immediately useful (eg genomics, bioimaging), KLD and MMD are still in wide use. (For example, our approach could potentially be applied to one of the MMD's most popular current applications -- two-sample testing -- in the presence of confounding.) Third, when non-identifiability leads to multiple solutions which provide equal fit to the data, minimal transport cost is an excellent "prior" but not the only defensible choice. L_p regularization, empirical Bayes weight sharing (used by ComBat), constraints (eg non-negativity or zero-off-diagonal) may instead be preferred, and may be fruitfully combined with KLD and/or MMD. Fourth, ConDo Wasserstein offers theoretical analysis opportunities which are distinctly different from the focus of this paper. We hope future papers will explore all these avenues.

Third, we agree that the restriction to affine transformations is limiting. But, our framework allows one to easily "plug in" more complex transformations. And our provided software, using PyTorch, makes future implementation of such extensions quite easy. So we hope future papers explore such opportunities.

---

### Decision · Action_Editors · 2022-06-10

**Recommendation:** Reject

**Comment:**

This paper proposes a novel DA procedure that consists in estimating a linear
mapping between two domains and to use a confounding variable to estimate this
mapping properly in the presence of both covariate and target shift. The authors
introduce it very simply in the introduction where a V1 domain has access to
samples $x^1_i,y^1_i,u^1_i$ where $x_i$ are features, $y_i$ are labels (that we
want to predict and $u_i$ are the confounding variable and access in the V2
domain to $x^2_i,u^2_i$ where labels $y$ are not available. This is a new way to
model data that can give better way to model a mapping between two domains and
the authors propose to estimate a linear or affine mapping my minimize the
expectation (wrt u) of a divergence of the conditional distributions of features
wrt u. The authors discuss different way to estimate the confounder prior
distribution and the conditional distributions, and present two divergences to
 optimize ! reverse KL and MMD. Numerical experiments illustrate the interest of
the method on a 1D and 2D synthetic dataset with continuous and discrete
confounder, on an image color grading illustration and one a unique Gene
Expression Batch Effect Correction real life dataset.

The idea of a using confounder to estimate a mapping between domains is elegant and  novel and
the reviewers found that the method is of interest to the community and
deserves to be published  but the consensus was that the paper is hard to follow
even for expert in DA and should undergo a major revision before being published
in a journal such as TMLR. For this reason the final decision is a rejection
with a strong recommendation to resubmit after taking into account the comments
in the reviews and those provided below.


Detailed comments:

+ The authors present the data as V1 V2 and then call V2, that is the domain
  where one wants to predict $y$ on new data, the "source domain" and V1 the
  target domain. This is a highly non-standard denotation in DA that the authors
  justify because they estimate a mapping from V2 to V1 since a good predictor
  is supposedly estimated in V1. But this notation is non-standard and makes the
  paper harder to understand for most readers (as it did for the reviewers and
  the AC). When using standard source/target domains the mapping estimated goes
  from target to source which is already well denoted in the title of the paper
  with "Backwards-Compatible" and the authors should stick to standard
  notations.

+ As discussed with the reviewers, the positioning with respect to the state of
  the art was a bit lacking and some references needed to be added. It has been
  done in the reviewed version but remain unclear. It deserve a proper section
  either in the introduction (where only one reference is used) or in section 2
  instead of the discussion below equation 4 and at the end of the paper. Also
  note that the use of Gaussian OT mapping for DA has been proposed (and
  studied) to the best of my knowledge in [A] where the mapping from V2 to V1 is
  learned with OT between Gaussians. Learning a mapping between domains with OT
  was originally proposed in [B] with estimation of an OT based non linear
  mapping from V1 to V2. Note that I am aware that I am co-author of these
  references but I honestly believe they are relevant in this case. Those
  missing references did not have an impact on the final decision.

+ The choice of the authors to not use a confounding variable $u$ but instead
  directly the target variable $y=u$ in the whole section 3 (after introducing
  all of it in Figure 1) makes the paper harder to read and understand. In
  unsupervised DA that is the standard DA problem (studied in most of the
  references in the paper) there is no target label available in the V2 domain
  so the method as described in section 3 cannot be applied. But indeed there
  might be access to a confounding variable that can be used to estimate the
  mapping. The authors should keep the more general problem in section 3 and
  discuss the special case $u=y$ in a paragraph or subsection. Note that if
  $u=y$ and labels are available in V2  then the problem can be cast as a
  multi-task learning problem instead of DA and for which there exists numerous
  exiting approaches.

+ This special case $u=y$ raises a lot of questions that should be discussed in
  the paper. If target labels are available in V2 why not train directly on this
  data? the authors did a partial response to that in the reply to reviewers
  (not enough data to train well in V2, need to remain on V1) but this should be
  investigated in the numerical experiments.  Note that training  a good
  classifier in V1 cannot compensate for a badly estimated mapping due to a
  small number of samples in V2. As a matter of fact for Gaussian OT DA, the
  generalization bound from [A] is in $O(max(n_1^{-1/2}, n_2^{-1/2}))$ that is
  for a small number of samples $n_2$ in V2 the same as supervised training
  generalization on V2 that would be $O(n_2^{-1/2})$.

+ The numerical experiments are convincing but limited to toy data and one real
  life dataset (color grading is closer to toy data). As such it is not a
  problem but the authors give the reader several large figures with large
  number of plots that are particularly hard to follow. It would make the paper
  easier to read if the authors illustrate the different DA problems
  investigated on a few examples and provide a visual comparison (barplot with
  error bars?) easier to interpret to interpret form than grids of plots and
  tables. Small comment : since in the toy data the actual mapping is known it
  would be interesting to measure the accuracy of its estimation directly
  (expected error in L2 sens). This also raises the question of when the method
  works best in practice (wrt the size of V1/V2) that would be interesting to
  investigate.


[A] Flamary, R., Lounici, K., & Ferrari, A. (2019). Concentration bounds for
linear Monge mapping estimation and optimal transport domain adaptation. arXiv
preprint arXiv:1905.10155.

[B] Courty, N., Flamary, R., & Tuia, D. (2014, September). Domain adaptation
with regularized optimal transport. In Joint European Conference on Machine
Learning and Knowledge Discovery in Databases (pp. 274-289). Springer, Berlin,
Heidelberg.